# ROVER: Recursive Reasoning Over Videos with Vision-Language Models for Embodied Tasks

Philip Schroeder[1]    Ondrej Biza[2]    Thomas Weng[2]    Hongyin Luo[1]    James Glass[1]

[1]MIT CSAIL      [2]RAI Institute

pschro@mit.edu

## Abstract

Vision-language models (VLMs) have exhibited impressive capabilities across diverse image understanding tasks, but still struggle in settings that require reasoning over extended sequences of camera frames from a video. This limits their utility in embodied settings, which require reasoning over long frame sequences from a continuous stream of visual input at each moment of a task attempt. To address this limitation, we propose ROVER (Reasoning Over VidEo Recursively), a framework that enables the model to recursively decompose long-horizon video trajectories into segments corresponding to shorter subtasks within the trajectory. In doing so, ROVER facilitates more focused and accurate reasoning over temporally localized frame sequences without losing global context. We evaluate ROVER, implemented using an in-context learning approach, on diverse OpenX Embodiment videos and on a new dataset derived from RoboCasa that consists of 543 videos showing both expert and perturbed non-expert trajectories across 27 robotic manipulation tasks. ROVER outperforms strong baselines across three video reasoning tasks: task progress estimation, frame-level natural language reasoning, and video question answering. We observe that, by reducing the number of frames the model reasons over at each timestep, ROVER mitigates hallucinations, especially during unexpected or non-optimal moments of a trajectory. In addition, by enabling the implementation of a subtask-specific sliding context window, ROVER's time complexity scales linearly with video length, an asymptotic improvement over baselines. Demos, code, and data available at: https://rover-vlm.github.io

## 1   Introduction

Vision-language models (VLMs) have demonstrated remarkable generalization capabilities across a wide range of tasks, including image captioning, visual question answering, and grounding language in complex visual scenes [3, 22, 52]. These models excel at extracting high-level semantic understanding from visual inputs and can generate fluent natural language that reflects this understanding [18, 56]. As a result, VLMs have become a key component in efforts to build general-purpose AI systems.

Given their success, VLMs are increasingly viewed as promising foundations for embodied intelligence: systems that perceive, reason about, and act within the physical world [38, 9, 19, 15, 31, 57, 25, 54, 63]. Embodied tasks require VLMs to recognize and reason about what is happening at each moment in time (given a continuous stream of visual input from camera video) when attempting to complete a task. Existing frameworks for VLM reasoning in this setting either perform highly localized reasoning over a small set of frames [57, 25, 54, 63] or attempt to reason over the entire sequence by concatenating all video frames into a single context [31]. Both strategies have limitations: the former sacrifices global context, while the latter is computationally expensive and overwhelms the model with irrelevant or redundant visual input. In this work, we seek to advance VLM capabilities in embodied settings by enabling high-precision frame-by-frame reasoning, without compromising efficiency or losing global context for the full task trajectory. With this motivation, we propose **R**easoning **O**ver **V**id**E**o **R**ecursively (ROVER).

39th Conference on Neural Information Processing Systems (NeurIPS 2025).

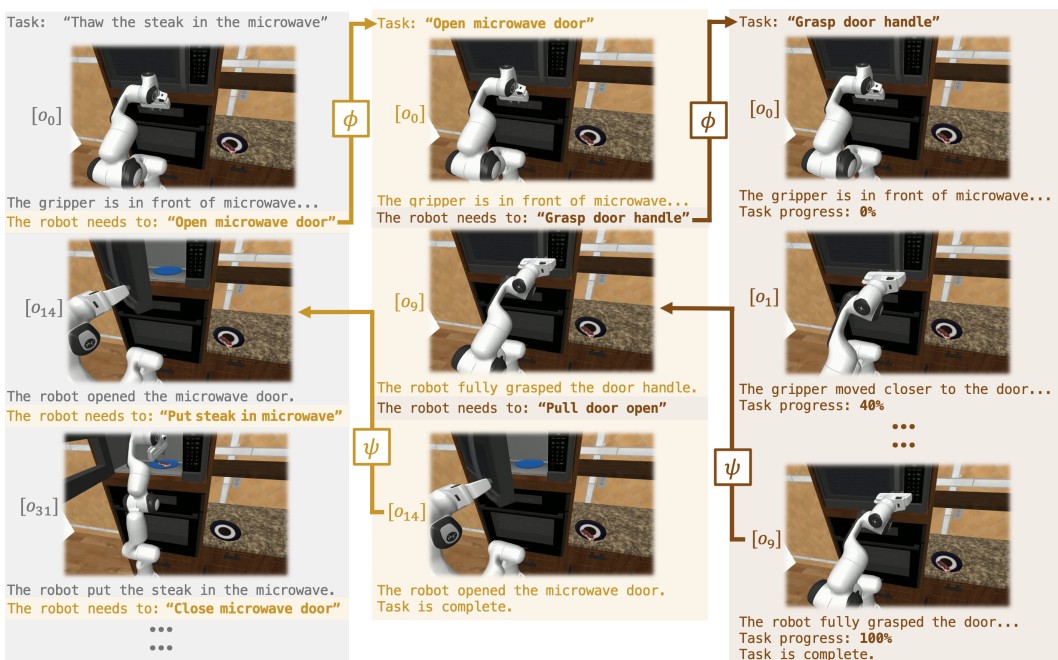

Figure 1: ROVER is a recursive framework for reasoning over camera video that decomposes a task into subtasks to maintain a compact temporal context, improving reasoning accuracy and efficiency.

ROVER is a framework that enables the model to recursively decompose the task corresponding to the given video input, reducing the number of camera frames the VLM must reason over at any given moment during the trajectory (Figure 1). Instead of generating a long, single line of reasoning spanning all timesteps of the video input for a task attempt (e.g., opening a door), ROVER decomposes the task and generates a separate line of reasoning for each subtask (e.g., grasping the door handle). When a subtask is complete, the corresponding line of reasoning terminates and a new one is created for the video input for the next subtask (e.g., pulling the door open). We show that the decomposition not only improves accuracy by focusing the reasoning on relevant temporal segments, but also enables the implementation of a subtask-specific sliding context window, which further reduces the number of frames the model must reason over at each moment of a trajectory.

We evaluate ROVER, implemented using an in-context learning approach, in the setting of robotic manipulation tasks using a large-scale dataset of videos collected from robot-mounted and third-person camera viewpoints during both successful and unsuccessful task attempts. We create this dataset by automatically perturbing expert demonstrations collected in RoboCasa [39] to produce diverse trajectories, ranging from near-optimal to fully random action sequences. In addition, we compute ground-truth task progress estimates at each timestep based on geometric distance to goal states. The generated dataset includes 543 videos across 27 tasks, each collected in a random kitchen scene. We leverage this dataset to evaluate ROVER across three benchmarks for embodied reasoning over camera video: 1) frame-level task progress estimation, 2) frame-level natural language reasoning, and 3) video question answering (QA). We also test ROVER on previous benchmarks based on diverse real-world OpenX Embodiment videos [41].

ROVER outperforms baselines across all settings: it achieves a stronger correlation with ground-truth task progress, lower frame-level reasoning error rates, and higher video QA accuracy. Specifically, we observe that ROVER reduces model hallucinations, especially during moments of a trajectory that deviate from expected or optimal behavior. We show these types of hallucinations during non-optimal states are most likely to occur when the model is reasoning over long sequences of camera frames. ROVER mitigates these hallucinations by reducing the number of frames the model must reason over at each timestep. In addition, by leveraging a sliding window within each subtask, ROVER's time complexity scales linearly with video length, an asymptotic improvement over baselines.
Overall, the contributions of this work including the following:

- We propose ROVER, a recursive framework that improves video reasoning accuracy and efficiency in embodied settings.

- We introduce a protocol to generate diverse non-expert trajectories with ground-truth progress labels from expert demonstrations in simulation.
- We release a large-scale video dataset featuring both expert and non-expert robotic task executions, annotated with natural language descriptions and fine-grained task progress signals.
- We establish a video reasoning benchmark that evaluates models on frame-level progress estimation, frame-level natural language reasoning, and video QA across diverse task trajectories.

## 2  Problem setup

We model tasks as goal-conditioned finite-horizon partially observed Markov decision processes:

$$\mathcal{M} := \langle S, A, P, R, \Omega, O, T, G \rangle. \tag{1}$$

The tuple $\langle S, A, P, R, G \rangle$ forms a Markov Decision Process, where $S$ is the (unobserved) state space, $A$ is the action space, $P : S \times A \to \Delta(S)$ is the state-transition function and $R : S \times G \to \mathbb{R}$ is the reward function. An agent observes the world through $O : S \to \Delta(\Omega)$, the observation function, where $\Omega$ is the set of observations. Finally, $T$ is the time horizon and $G$ specifies the task semantically. Conditioned on a history $h_t = (a_{<t}, o_{\leq t}) \in H$ and a goal $g \in G$, we aim to find a policy $\pi : H \times G \to A$ that maximizes the $\gamma$-discounted return $J_\pi = \mathbb{E}_\pi \left[ \sum_{t=0}^{T-1} \gamma^t R(s_t, a_t, g) \right]$. The goal-conditioned value function $V^\pi$ is defined as the expected cumulative reward when starting from an initial state $s_0$ and following policy $\pi$:

$$V^\pi(s_0; g) = \mathbb{E}_{\pi, P} \left[ \sum_{t=0}^{T-1} \gamma^t R(s_t, a_t, g) \,\middle|\, s_0 \right].$$

Given a task description $g \in G$ and sequence of observations $\{o_t\}_{t=1}^T$ from video input, our goal is to generate text, $\{m_t\}_{t=1}^T$, following each observation that reasons in natural language about the task trajectory relative to the goal $g$, including estimating task progress, at every timestep. Section 3 outlines the recursive framework we propose for this reasoning task. To evaluate reasoning accuracy, we generate a diverse dataset of videos, $\mathcal{D}$, exhibiting task attempts with varying levels of expertise by inserting random perturbations within a given dataset, $\mathcal{D}_{src}$, of expert trajectories in simulation (described in section 4.1). In addition, we compute ground-truth task progress estimates at each timestep of the generated videos based on geometric distance to goal states (described in section 4.2).

## 3  ROVER framework for reasoning over video trajectories

Given a task description $g$ and visual input $\Omega$ from camera video, ROVER recursively generates sequences of reasoning at each timestep of the task attempt. The sequences of reasoning are comprised of camera frames for each timestep, interwoven with natural language text (Figure 1). ROVER can be implemented using the recursive function shown in Algorithm 1. The function takes two inputs: the context $c$ for the given task, and the sequence of tokens, $Y$, that have been provided or generated so far for this task. $Y$ includes text tokens generated by the VLM and image tokens representing the frames from the camera input. We treat a token sequence (e.g., $c$ and $Y$) as a list of tokens.

ROVER begins with $Y = [\,]$ and $c$ includes text tokens of the initial task description (e.g., "Thaw the steak in the microwave") and the first observation (image tokens of the camera frame). The model generates, $\Theta(c + Y)$, one token at a time conditioning on the given context, $c$, appended with the growing token sequence, $Y$.

If $Y$ includes the tokens for retrieving the next observation (details explained below), then the image tokens for the camera frame at the next timestep, $o_{next}$, are appended to $Y$. If $Y$ specifies a new subtask, then a new line of reasoning is created with $\text{ROVER}(\phi(Y), [\,])$ where $Y$ for the child process is initialized as an empty list

---

**Algorithm 1** ROVER

1: **function** ROVER($c, Y$)
2:     **while** True **do**
3:         $Y = Y + \Theta(c + Y)$
4:         **if** $Y$ progresses to next frame **then**
5:             $Y = Y + o_{next}$
6:         **else if** $Y$ specifies a new task **then**
7:             $Y = Y + \psi(\,\text{ROVER}(\phi(Y), [\,])\,)$
8:         **else if** $Y$ indicates task complete **then**
9:             **return** $Y$

---

and $c$ is based on the token sequence of the parent, $\phi(Y)$. When the child process ends, its output tokens are appended to the parent's token sequence and the parent continues generating. If $Y$ indicates the task is complete, or the end of the video is reached, then the process returns $Y$ (its full token sequence).

**Implementation of $\phi$ and $\psi$ functions.** The $\phi$ function defines the context for a child process based on the full token sequence of the parent at the time the child is spawned, including tokens directly generated by the parent or returned as output from previous child processes of that parent. The $\psi$ function defines the output tokens of a child based on its full token sequence at the time it terminates. In our implementation of ROVER (as depicted in Figure 1), $\phi$ returns the new task description provided by the parent along with the last frame of the parent line of reasoning. We implement $\psi$ as a function that returns the final frame, and corresponding final text description, of the child process.

**Special text tokens for retrieving next frame and specifying new subtasks.** When the VLM is done generating the natural language text tokens for reasoning about the current camera frame, it either moves on to the frame for the next timestep or specifies a new subtask. The next frame of the video can be retrieved by generating the text tokens "[next-frame]". A new subtask can be specified by generating the text tokens "The robot needs to: {new_subtask}", as depicted in Figure 1.

**Sliding context window for video.** In our implementation of ROVER, we find that using a sliding window at each timestep not only improves inference efficiency, but also mitigates observed problems with hallucination. When appending a new frame, $Y = \text{Window}(Y + o_{next})$, the window extracts three frames: the first frame of the current line of reasoning and the most recent previous frame (each paired with their previously generated text descriptions), along with the newly added next frame. We observe a complementary effect between the sliding window and the recursive decomposition. The window forces the model to focus on the most relevant frames for the current subtask and the current timestep, while the recursive specification of new subtasks ensures the current subtask description and first frame of the subtask-specific window are continually updated as task milestones are achieved during the video. Further, by capping the maximum number of frames included in the context to three, the inference time complexity of ROVER scales linearly with the total video frame count. This provides an asymptotic improvement over baselines, which scale quadratically with total frame count.

## 4 Generating diverse video trajectories and ground-truth value estimates

To test vision-language reasoning across diverse video trajectories, we seek to use a source dataset of expert demonstrations, $\mathcal{D}_{src}$, for a task in simulation to build a dataset of trajectories, $\mathcal{D}$, exhibiting a wide spectrum of expertise with that task, along with value estimates for all states. We focus on robot manipulation tasks and, following prior work [36, 11], we assume tasks consist of a known sequence of object-centric subtasks. If we let $E = \{e_1, e_2, ...\}$ be the set of entities (objects) in a task $\mathcal{M}$, we assume that tasks consist of a sequence of object-centric subtask trajectories $\tau = \{\tau_i(e_{\tau_i})\}_{i=1}^M$, where the manipulation in each subtask trajectory $\tau_i(e_{\tau_i})$ is relative to a single object's coordinate frame ($e_{\tau_i} \in E$). We include details regarding how this sequence is specified for each task in Appendix M.

### 4.1 Non-expert trajectory generation

In this section, we outline an approach which, given a set of expert demonstrations, $\mathcal{D}_{src}$, generates a dataset of trajectories, $\mathcal{D}$, exhibiting varying levels of expertise, ranging from nearly expert to fully random action sequences. Each expert demonstration, $\tau^E \in \mathcal{D}_{src}$, is decomposed into object-centric segments $\tau^E = \{\tau_i^E(e_{\tau_i})\}_{i=1}^M$ (see Appendix M for full details). Following MimicGen [36], each subtask trajectory $\tau_i^E = \{(s_t^E, a_t^E)\}_{t=1}^{T_i^E}$ corresponds to a manipulation relative to a single object $e_{\tau_i}$.

Non-expert trajectories are derived by injecting random deviations into these expert demonstrations (Figure 2). For a given subtask trajectory $\tau_i^E$, we construct a corresponding non-expert version $\tau_i^N = \{(s_t^N, a_t^N)\}_{t=1}^{T_i^N}$ by inserting random actions at selected moments $q \in \mathcal{Q}_i \subseteq \{1, \ldots, T_i^E\}$. Each time in $\mathcal{Q}_i$ marks a branching point, at which $\tau_i^N$ diverges from $\tau_i^E$ for some number of steps, $w$. The index of each step in the deviation is specified based on the index of the expert state, $q$, at which the deviation began, along with the number of steps, $j$ (ranging from 0 to $w-1$), it has deviated from that expert state. For each step during the deviation $\{q, j\}_{j=0}^{w-1}$, a randomly sampled action $a_{q,j}^N \sim \mathcal{U}(A)$ is executed, where $\mathcal{U}(A)$ denotes the uniform distribution over the action space.

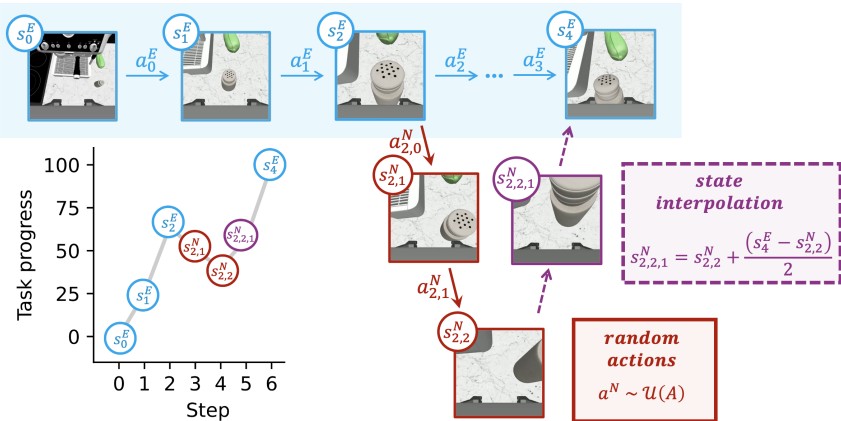

Figure 2: Generating diverse video trajectories by inserting random deviations during expert path.

The deviation leads to a sequence of states and actions $\{s_{q,j}, a_{q,j}\}_{j=0}^{w-1}$ that diverges from the expert trajectory, where $s_{q,0}$ is the expert state from which the deviation began. Following deviation, the trajectory can either terminate or start to recover to the expert trajectory. Recovery is implemented by interpolating between the deviated state and a downstream expert state (Figure 2). Specifically, given the final deviated state $s_{q,w}^N$ and a selected future expert state $s_{q+h}^E$ (for some $h > 0$), we generate a sequence of interpolated states:

$$s_{q,w,z}^N = (1 - \alpha_z) \cdot s_{q,w}^N + \alpha_z \cdot s_{q+h}^E, \quad \alpha_z = \frac{z}{n_{\text{interp}}}, \quad z = 1, \ldots, n_{\text{interp}},$$

to gradually transition back to the expert trajectory. The index of each step during recovery is specified based on the index of the expert state, $q$, at which the original deviation began, the deviation length, $w$, and the number of interpolation steps, $z$ (from 1 to $n_{\text{interp}}$) that have occurred. $n_{\text{interp}}$ determines the smoothness of the recovery. This procedure results in trajectories that deviate to varying extents from the expert trajectory. We ensure diversity across trajectories by varying deviation timesteps, number and magnitude of deviations (including nested deviations), and recovery points.

## 4.2 Value functions

For each task, we define a ground-truth value function to evaluate the accuracy of VLM reasoning throughout the video trajectories. In robotics, value functions can be difficult to define due to task heterogeneity. We follow prior work [14, 29, 31, 45, 46, 53] in modeling value using task progress, which provides a normalized temporal measure of how far an agent has advanced toward completing a task. Formally, we define a progress-based value function, $V : \mathcal{S} \times G \to [0, 1]$, mapping an observation and goal specification to a scalar value between 0 and 1.

**Goal-focused distance measures.** To quantify progress, we define subtask-specific metrics based on spatial relationships in the observable state. For each subtask, we identify key geometric distances that reflect meaningful aspects of completion and are continuous and monotonic with respect to progress. These include the sum of distances between the contact points of the robot end effector, $\{r_1, r_2, \cdots, r_C\}$, and the corresponding contact points of the target object, $\{l_1, l_2, \cdots, l_C\}$, in the scene, denoted $y^{r,e}$, along with the distance between the current position of the target object and its respective goal position, denoted $y_t^{e,f}$,

$$y_t^{r,e} = \sum_{j=1}^{C} \left\| p_t^{r_j} - p_t^{l_j} \right\|_2, \quad y_t^{e,f} = \left\| p_t^{e_{\tau_i}} - p_f^{e_{\tau_i}} \right\|_2$$

where $e_{\tau_i}$ is the subtask-specific target object discussed above, $p_t^{r_j}$ and $p_t^{l_j}$ are positions of the $j^{th}$ robot and object contact point, respectively, at time $t$, $p_t^{e_{\tau_i}}$ is the position of $e_{\tau_i}$ at time $t$, and $p_f^{e_{\tau_i}}$ is the target position of $e_{\tau_i}$. All distances are computed using the L2 norm (Euclidean distance). The environment includes $B$ entities, each of which is represented by a position in three dimensional Cartesian space within a global coordinate frame. The total goal-focused distance, $y = \{y_t\}_{t=1}^{T_i}$, is

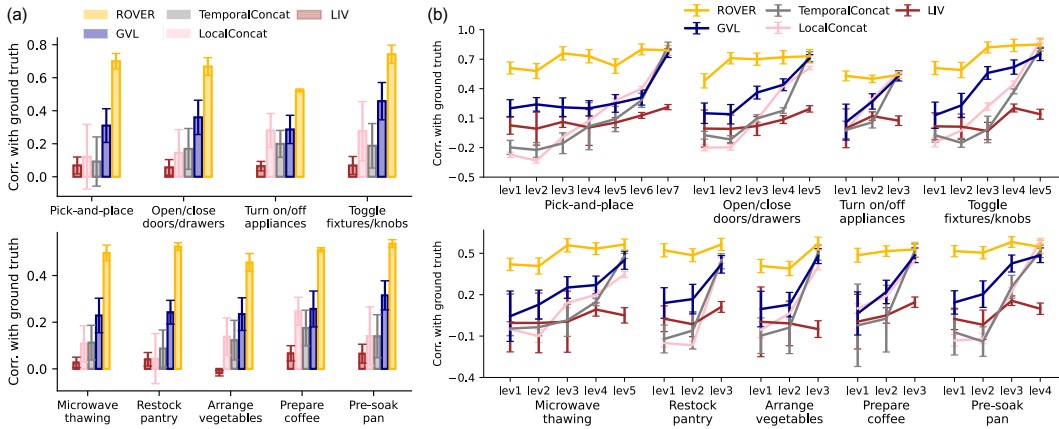

Figure 3: Mean and standard error of correlation between ground-truth values and progress values predicted for all videos (a) and stratified across trajectory level (b). Highest-level videos for each task show near-expert completion. Amount of non-expert behavior increases as level decreases.

the sum of these metrics, $y_t = (1 - \beta)\, y_t^{r,e} + \beta\, y_t^{e,f}$, where $\beta$ is a scalar with range $[0, 1]$ that is manually tuned for each object-centric subtask to ensure the degree to which $y^{r,e}$ and $y^{e,f}$ contribute to $y$ is proportional to how meaningful each component is to the overall subtask completion.

**Trajectory-focused distance measures.** The goal-focused metrics are necessary to track what we value in the task (e.g., moving the gripper closer to the target object, the target object moving closer to the goal position). However, they may fail to capture important nuances in real-world trajectories. For instance, the shortest Euclidean path between the robot gripper and target objects may intersect with obstacles that must be avoided. To address this, we refine the value function by leveraging known expert states within each trajectory. Within the expert demonstrations, we identify a sparse set of keypoints, $\kappa = \{s_{k_i}^E\}_{i=1}^n$, that form the local maxima in the goal-focused distance, $y = \{y_t\}_{t=1}^T$, for a given object-centric subtask. Since these keypoints derive from the expert demonstrations, we know that, despite forming local maxima in $y$, they represent intermediate states that lie along high-value trajectories. For each timestep in a non-expert trajectory, we identify the nearest downstream expert keypoint and use this as an auxiliary signal to adjust the local value estimate.

$$u_t = \|s_{t,j,z}^N - s_{k_\text{next}}^E\|_2, \quad k_\text{next} = \min\{k_i \in \kappa \mid k_i > t\}, \quad \text{where } j = z = 0 \text{ when } t \notin \mathcal{Q}_i$$

where $s_{t,j,z}^N$ is the environment state at the current step of the non-expert trajectory and $s_{k_\text{next}}^E \in \mathbb{R}^{3 \times B}$ is the full environment state at time $k_\text{next}$ during the expert trajectory.

**Final value estimates.** We transform the raw distance measures for the full subtask trajectory into normalized progress values, $v = \{v_t\}_{t=1}^{T_i}$, between 0 and 1 by inverting and rescaling them:

$$v_t = \frac{-d_t + \max(d)}{-\min(d) + \max(d)}, \quad d_t = y_t + u_t, \quad d = \{d_t\}_{t=1}^{T_i}$$

To form the value estimates for the full trajectory, $\tau = \{\tau_i(e_{\tau_i})\}_{i=1}^M$, we sequence together the value estimates corresponding to each subtask, adding the value estimates of each subtask to the final value of the previous subtask and, when complete, rescaling the list of values to the range $[0, 1]$.

## 5 Experiments

We conduct large-scale experiments to evaluate ROVER across diverse tasks for embodied reasoning over camera video. The baseline methods include the state-of-the-art value model LIV [32], a multi-modal contrastive model [43] fine-tuned with a value learning objective. We also include the best existing in-context learning approach, Generative Value Learning (GVL) [31], and an ablated version of GVL, called TemporalConcat, that processes frames in temporal order. Further, we include a local reasoning baseline, LocalConcat, that processes frames in temporal order, but only includes the last three frames of the video in the model's context at each timestep. Similar to GVL, we implement ROVER using an in-context learning approach, with the same prompting used for all backbone VLMs (details in Appendix D). We test ROVER using various backbone models. The main results are

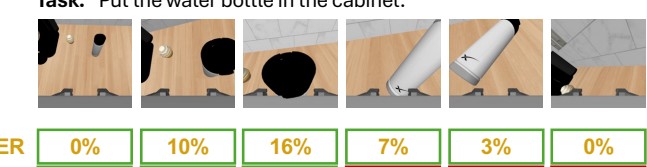

**Task:** "Put the water bottle in the cabinet."

| | | | | | | |
|---|---|---|---|---|---|---|
| **ROVER** | 0% | 10% | 16% | 7% | 3% | 0% |
| **GVL** | 0% | 12% | 18% | 15% | 34% | 23% |
| *TemporalConcat* | 0% | 8% | 21% | 44% | 62% | 87% |
| **Ground truth** | 0% | 9% | 17% | 8% | 5% | 3% |

Figure 4: ROVER exhibits more accurate reasoning and progress prediction during non-expert states.

based on Gemini-1.5-Pro. Appendix J includes results for Gemini-2.5-Pro-Preview, GPT-4o, and Qwen-2.5-VL-32B-Instruct. Examples showcasing ROVER video reasoning, along with anonymous links to all code and data, are provided at: `https://rover-vlm.github.io/anon/`

We use RoboCasa [39] as the simulation environment and leverage the demonstrations they provide (collected via human teleoperation) as the source dataset of expert trajectories (details in Appendix E). Our generated evaluation dataset, comprising trajectories that exhibit a wide range of task expertise, includes 543 videos across 27 tasks (Appendix B), each collected in a random kitchen scene. The videos are separated into levels based on the amount of the task completed during the video (Appendix B). The highest-level videos in each task group show full task completion with near-expert behavior.

### 5.1 Vision-language reasoning tasks with generated video dataset

Our simulation-generated video trajectories, each annotated with dense value estimates, enable the evaluation of multiple vision-language reasoning tasks that test fine-grained temporal understanding and visual grounding. We separate the evaluation into the following three tasks: 1) frame-level task progress estimation, 2) frame-level natural language reasoning, and 3) video question answering (QA). Further details regarding the task design and performance metrics are provided in Appendix C.

### 5.2 Task 1: Frame-level task progress estimation

This benchmark involves predicting task progress at each timestep of a video (see Appendix C.1 for details). Following prior work [31], we implement this by having the model predict integer values representing the percentage progress achieved relative to the first frame of the video, which is always assigned a progress value of zero. We implement ROVER such that the model predicts task progress values up to 100 for each subtask, as depicted in Figure 1. To form progress values for the full task horizon, we divide the values within each subtask by the total number of subtasks and add them to the value for the last timestep of the previous subtask (see Appendix D for details).

**ROVER shows higher correlation with ground-truth task progress estimates.** For videos that exhibit task completion with near-expert behavior (i.e., the highest level within each task group), ROVER, GVL, TemporalConcat, and LocalConcat achieve a Pearson correlation near or above 0.5 (Figure 3) and an L2 distance below 200 (Appendix A.1), relative to the ground-truth value estimates, across most task groups. However, for videos with incomplete task execution, GVL, TemporalConcat, and LocalConcat deviate significantly from the ground truth (as depicted in Figure 4). These deviations become more extreme as the trajectory level decreases (i.e., as the number of non-expert states in the video increases).

**ROVER achieves higher correlation with frame number on real-world expert demonstrations.** In Appendix K, we include the same value estimation analysis conducted in [31] using 1,000 videos from 50 real-world datasets from OXE [41]. We see that ROVER achieves higher correlation with ground-truth value estimates (which are based on frame number) across the OXE datasets, most of which contain human-collected expert demonstrations.

### 5.3 Task 2: Frame-level natural language reasoning

At each frame of the video, ROVER, along with baselines, generates a description of the frame before predicting task progress for that timestep. We cross-reference these descriptions with information about the ground-truth simulator state at each timestep to calculate the reasoning error rate and

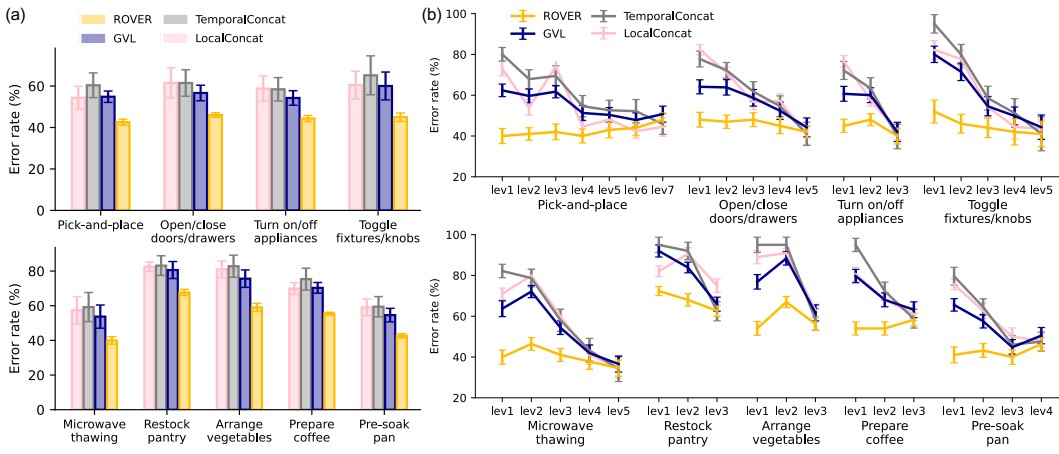

Figure 5: Mean and standard error of reasoning error rate (percentage of frames model states something that is verifiably wrong) for all videos (a) and stratified across trajectory level (b).

success rate (described in Appendix C.2). We treat this evaluation as a language model task, where we prompt the model with the generated description paired with the ground-truth information and ask if the description includes any statements that are verifiably true or false relative to the ground truth.

**ROVER achieves lower error rate in frame-level reasoning during video.** The reasoning error rate is similar for all methods for videos containing near-expert task completion (Figure 5). However, as videos deviates from expert behavior, the error rate increases significantly for GVL, TemporalConcat, and LocalConcat. We observe a similar pattern for reasoning success rate (Appendix A.2). These trends mirror the results of the progress prediction task, suggesting that errors in progress prediction result from errors in the natural language descriptions of the frame that precedes the progress prediction at each timestep. We further evaluate the nature of these errors in the video QA analysis below.

**ROVER reduces errors by focusing context.** When stratifying results across state type (expert, non-expert, and interpolating states) and context length in Appendix H, we see that GVL, TemporalConcat, and LocalConcat are most likely to error when encountering non-expert states and when the number of frames included in the context extends beyond 10. We observe a similar pattern when stratifying the progress prediction results across state type (Appendix H). ROVER reduces these errors by using recursive decomposition, along with a sliding window, to narrow the context at each timestep.

## 5.4 Task 3: Video QA

The video QA task includes questions about whether specific events occur (and the time they occur) during the video (details in Appendix C.3). We generate answers to each question for this task using the frame-level descriptions produced by each method. For each question, we use a language model to generate an answer given all frame descriptions concatenated together. In doing so, the QA performance provides a secondary measure of the quality of the frame-level natural language reasoning of each method.

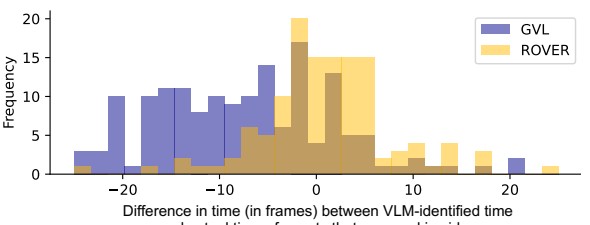

Figure 6: Difference in time (in frames) between VLM-identified time and actual time of events that occurred in video. One second in real time corresponds to ~3 frames. Negative frame difference indicates VLM states something occurs before it actually occurs during video.

**ROVER reduces hallucinations during videos exhibiting non-expert behavior.** ROVER shows significantly higher accuracy than GVL, TemporalConcat, and LocalConcat on the video QA benchmark across all task groups (Appendix A.3). The video QA results reveal a hallucination problem with GVL, TemporalConcat, and LocalConcat, where the VLM is likely to state that an event occurred during a video, regardless of whether it actually occurs. This is illustrated by the low precision (20 to 50%) and high recall values (near 100%) across all task groups for GVL, TemporalConcat, and

LocalConcat (Appendix A.3). The hallucination problem is also highlighted in the analysis of the distance between the time when something occurs in a video and the time when the VLM states that it occurred (Figure 6). We see that, even when GVL correctly states something occurs during a video, it is much more likely than ROVER to state this prematurely (as shown by the negative frame difference in Figure 6).

## 5.5 Error analysis

We hand-reviewed 100 real-world videos and 100 videos from simulation to identify the most common types of errors for each method (Table 1). In Appendix C.4, we outline the full list of error types and provide examples for each.

The most common error type for ROVER is specifying incorrect subtasks, where 9% of real-world videos and 7% of simulation videos include at least one occurrence of this error. The most common error type for GVL is perception errors, where 47% of real-world videos and 32% of simulation videos include at least one occurrence of this error. Further, we observe that 83% of perception errors from GVL occur when there are over 10 frames of the video included in the context. This supports our findings that the VLM is prone to perception errors, such as hallucination, when attempting to reason over many frames of a video. ROVER mitigates these perception errors by shortening the number of frames the model must reason over at each timestep of the video (through the synergistic combination of recursive decomposition and a sliding context window). Overall, we find that ROVER introduces the possibility of decomposition errors, but dramatically reduces the total error rate by mitigating the model's tendencies toward hallucination.

Table 1: Error rates (%) observed among 100 real-world videos and 100 videos from simulation

| Error type | Real-world videos | | Simulation videos | |
|---|---|---|---|---|
| | GVL | ROVER | GVL | ROVER |
| Incorrect subtasks | NA | 9% | NA | 7% |
| Redundant subtasks | NA | 3% | NA | 4% |
| Perception error | 47% | 7% | 32% | 5% |
| Reasoning error | 14% | 4% | 14% | 4% |
| Other | 15% | 3% | 11% | 3% |
| Total | 76% | 26% | 57% | 23% |

## 5.6 Ablations

We ablate ROVER to demonstrate how different algorithmic design decisions impact the performance gains we observe relative to baselines. Further, we show that ROVER is robust to video length and frame rate (Appendix H), camera view (Appendix I), and backbone VLM (Figure 7b, Appendix J).

We test two ablations of ROVER (details in Appendix L). One version ablates only the sliding window while the other ablates only the recursive reasoning. Consistent with prior work [31], we observe that baseline methods such as TemporalConcat often hallucinate steady, monotonically increasing task progress values, even when the video includes significant stalls or regressions. GVL mitigates this problem via random frame shuffling, which disrupts the strong temporal priors and forces the model to assess each frame more carefully [31]. We observe that the sliding window (even without recursion) similarly reduces hallucination by retaining temporal context while avoiding overly rigid sequential processing. Our ablations show that the window-only variant of ROVER significantly outperforms TemporalConcat and matches or exceeds GVL's performance on short-horizon tasks, such as turning on/off appliances (Figure 7a). We see that adding the recursive decomposition further improves performance for tasks consisting of numerous subtasks, such as pick-and-place tasks.

## 6 Related work

There is significant prior work leveraging VLMs in embodied reasoning settings [38, 9, 19, 15, 32]. Methods such as [31, 57, 25, 54, 63] have applied VLMs for reasoning over video sequences, either by attending to a small number of keyframes or processing the entire trajectory. However, these approaches struggle to balance fine-grained reasoning with the preservation of global task context. Separately, decomposition-based reasoning has been explored in both language and vision-language

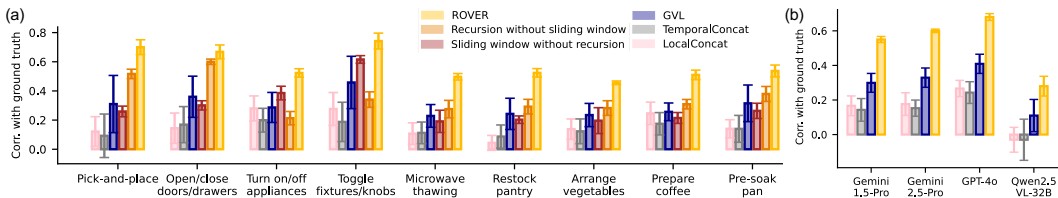

Figure 7: Mean and standard error of correlation between ground-truth value estimates and progress values predicted across all videos for (a) design ablations and (b) changing backbone VLM.

domains [24, 55, 16, 28, 37, 40, 64]. Recent work has also explored structured prompting and recursive task breaking in VLMs [21, 61, 42, 44], though primarily in static or text-centric settings. ROVER builds on these ideas by introducing a recursive, subtask-oriented video-reasoning framework to address challenges of continuous visual input in embodied tasks, allowing for more precise and scalable temporal reasoning without sacrificing global context. **More discussion in Appendix O.**

## 7 Limitations

ROVER improves reasoning over videos by recursively decomposing the given task into subtasks and, in doing so, maintaining a compact temporal context that mitigates hallucination. However, when decomposition fails (e.g., by introducing unnecessary or invalid subtasks), the resulting reasoning may become fragmented or misaligned with the actual task progression. In addition, we implement ROVER with an in-context learning approach. We leave to future work the evaluation of fine-tuning methods that can improve ROVER's performance with frame-level reasoning and value estimation.

## 8 Broader impact

In this work, we show how ROVER improves VLM reasoning in embodied settings. VLMs are being increasingly deployed to autonomously interact with external environments. By improving this capacity, our work has the potential for amplifying risk associated with automated decision-making. Addressing these risks requires careful consideration of ethical guidelines and robustness checks.

## 9 Conclusion

We present ROVER, a recursive framework that improves VLM reasoning over video sequences in embodied tasks. By decomposing complex tasks into subtasks and enabling localized, temporally bounded reasoning within each, ROVER effectively reduces cognitive load on the model and mitigates common failure modes such as hallucination during off-nominal behavior. Our experiments across three benchmarks (progress estimation, frame-level reasoning, and video QA), along with previous benchmarks based on diverse real-world OXE videos, demonstrate consistent improvements over strong baselines, particularly in handling diverse, suboptimal trajectories. In addition, by enabling the use of a subtask-specific sliding window, ROVER's time complexity scales linearly with video length, an asymptotic improvement over baselines. By combining architectural insights with a new large-scale dataset and evaluation protocol, our work lays a foundation for more robust and scalable embodied reasoning systems grounded in real-world video input.

## 10 Acknowledgements

This work was supported by the MIT-IBM Watson AI Lab and Quanta Computer, Inc.

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

## Appendix Contents

# A  Additional results for main experiments

## A.1  Task 1: Frame-level task progress estimation

**ROVER becomes less correlated with *frame number* as the number of non-expert states during video increases.** To further explore how reasoning quality changes as videos deviate from the expert behavior, we evaluate how the correlation between the video frame number and the predicted task progress changes as the number of non-expert states in the video increases. This provides a measure of reasoning quality that is independent of the ground-truth value estimates. The progress values generated by GVL and TemporalConcat show similar correlation with frame number regardless of the number of non-expert states in the video (Figure 8). In contrast, ROVER shows a significant drop in its correlation with frame number as the number of non-expert states in the video increases, as would be expected with accurate reasoning (and as is observed with the ground-truth value estimates).

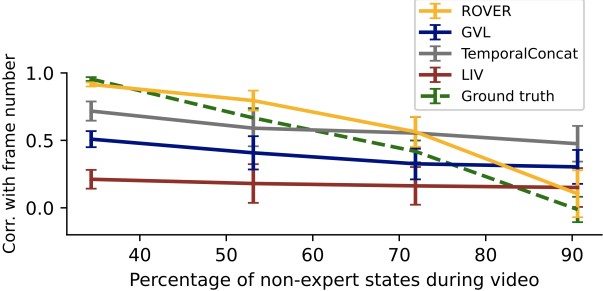

Figure 8: Mean and standard error of correlation between video frame number and the frame-level progress values predicted by each method stratified across proportion of non-expert states in video.

**ROVER achieves smaller distance between predicted and ground-truth progress values with non-expert trajectories.** Figure 9 is based on the same set of results presented in Figure 3, but uses L2 distance, instead of correlation, between model-predicted and ground-truth progress estimates as the performance metric. Similar to what we observe with the correlation results in Figure 3, we see in Figure 9 that ROVER significantly improves performance (i.e., better agreement with ground-truth value estimates reflected by smaller L2 distance) for videos exhibiting non-expert behavior. The performance improvement of ROVER over baselines becomes more significant as the trajectory level decreases (i.e., as the number of non-expert states in the video increases).

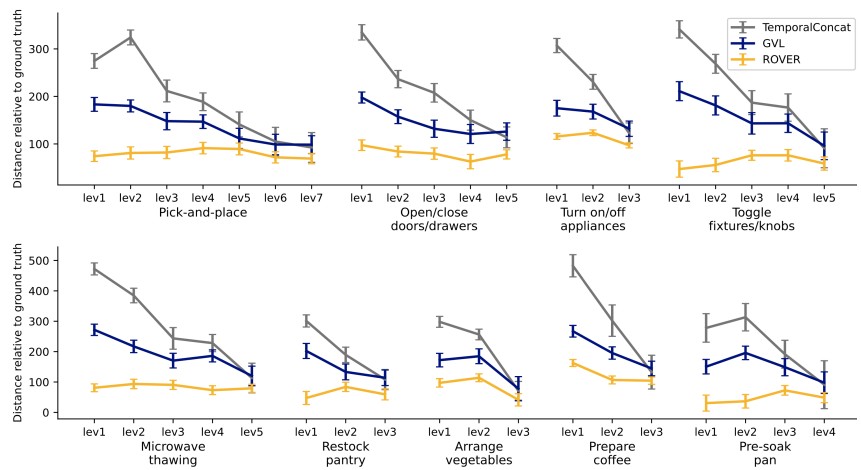

Figure 9: Mean and standard error of distance between model-predicted and ground-truth task progress estimates stratified across trajectory level.

## A.2 Task 2: Frame-level natural language reasoning

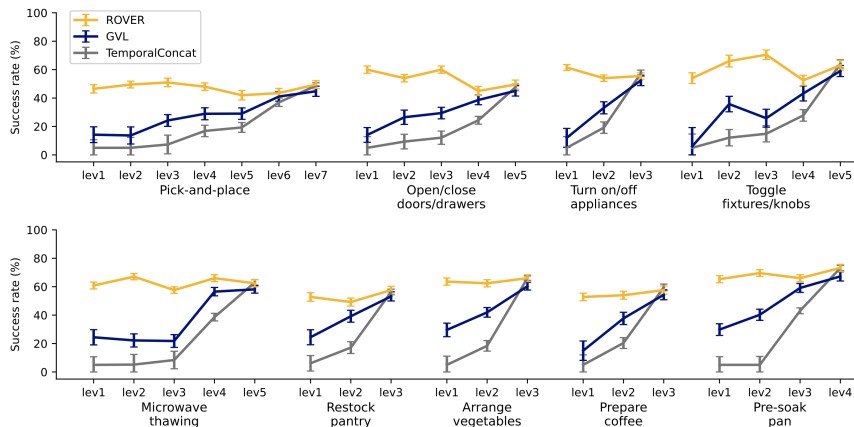

Figure 10: Mean and standard error of frame-level reasoning success rate (percentage of frames model states something that is verifiably correct relative to ground-truth information about simulator state) for videos stratified across trajectory level.

## A.3 Task 3: Video QA

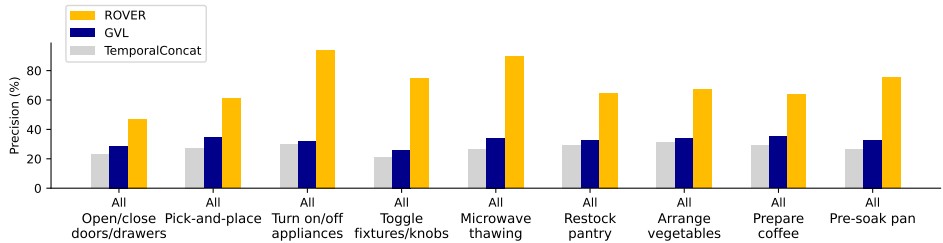

Figure 11: Video QA accuracy (the percentage of questions for which the model correctly identifies whether the event occurred) across all task groups.

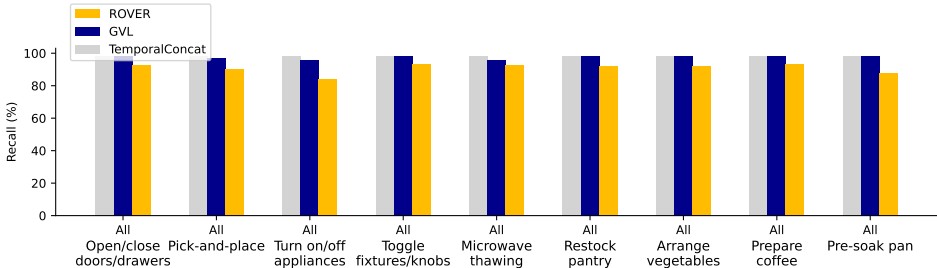

Figure 12: Video QA precision (the percentage of predicted occurrences that actually occurred in the video) across all task groups.

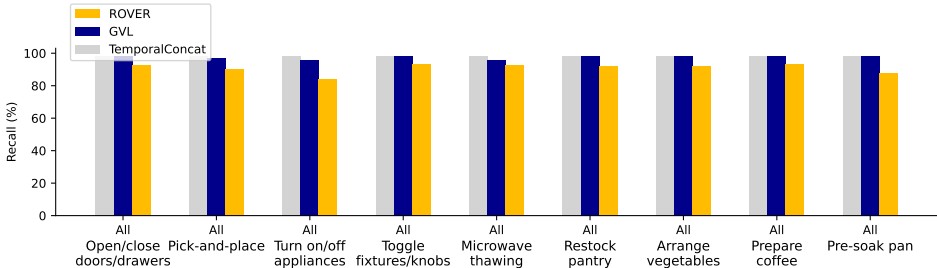

Figure 13: Video QA recall (the percentage of actual occurrences correctly identified by the model) across all task groups.

# B   Details on generated evaluation dataset

Our generated evaluation dataset comprises videos that exhibit a wide range of task expertise. The dataset includes 543 videos across 27 tasks, with about 20 videos per task (Tables 2 and 3). Each trajectory is collected in a random kitchen scene (random floor plan, random kitchen style, and random textures) in RoboCasa [39]. The videos are separated into levels based on the amount of the task completed during the video (Table 3). The highest-level videos in each task group show full task completion with near-expert behavior. The dataset contains both atomic and composite tasks from RoboCasa (descriptions of each task are provided in the appendix of [39]). The composite tasks involve two to four times as many actions as the atomic tasks. For evaluation, we downsample videos to 30 frames for the atomic tasks (following [31]) and donwsample videos to 60 frames for the longer composite tasks.

Table 2: Tasks within each task group.

| Task group | Task | Task type |
|---|---|---|
| Pick-and-place | PickPlaceCabToCounter | Atomic |
| | PickPlaceCounterToCab | |
| | PickPlaceCounterToMicrowave | |
| | PickPlaceCounterToSink | |
| | PickPlaceCounterToStove | |
| | PickPlaceMicrowaveToCounter | |
| | PickPlaceSinkToCounter | |
| | PickPlaceStoveToCounter | |
| | CoffeeSetupMug | |
| | CoffeeServeMug | |
| Open/close doors/drawers | OpenSingleDoor | Atomic |
| | CloseSingleDoor | |
| | OpenDrawer | |
| | CloseDrawer | |
| Turn on/off appliances | TurnOnMicrowave | Atomic |
| | TurnOffMicrowave | |
| | CoffeePressButton | |
| Toggle fixtures/knobs | TurnSinkSpout | Atomic |
| | TurnOnSinkFaucet | |
| | TurnOffSinkFaucet | |
| | TurnOnStove | |
| | TurnOffStove | |
| Microwave thawing | MicrowaveThawing | Composite |
| Restock pantry | RestockPantry | Composite |
| Arrange vegetables | ArrangeVegetables | Composite |
| Prepare coffee | PrepareCoffee | Composite |
| Pre-soak pan | PreSoakPan | Composite |

Table 3: Levels for non-expert trajectories and video dataset counts. For each level, we assume that all task components corresponding to the preceding levels have already been successfully completed.

| Task group | Non-expert trajectory levels | Dataset count |
|---|---|---|
| Pick-and-place | 1: Fail to approach {obj} | 3 |
| | 2: Approach {obj} | 3 |
| | 3: Contact {obj} | 3 |
| | 4: Pick up {obj} | 3 |
| | 5: Keep grasp of {obj} | 3 |
| | 6: Approach placing {obj} in {target_location} | 3 |
| | 7: Place {obj} in {target_location} | 3 |
| Open/close doors/drawers | 1: Fail to approach the door/drawer | 4 |
| | 2: Approach door/drawer | 4 |
| | 3: Contact door/drawer | 4 |
| | 4: Start opening/closing door/drawer | 4 |
| | 5: Finish opening/closing door/drawer | 4 |
| Turn on/off appliances | 1: Fail to approach the on/off button/lever | 6 |
| | 2: Approach the on/off button/lever | 6 |
| | 3: Successfully adjust the on/off button/lever | 6 |
| Toggle fixtures/knobs | 1: Fail to approach the lever/knob | 4 |
| | 2: Approach lever/knob | 4 |
| | 3: Contact lever/knob | 4 |
| | 4: Start turning/twisting lever/knob | 4 |
| | 5: Finish turning/twisting lever/knob | 4 |
| Microwave thawing | 1: Fail to open microwave door | 4 |
| | 2: Open microwave door | 4 |
| | 3: Pick up the food item and place it in the microwave | 4 |
| | 4: Close the microwave door | 4 |
| | 5: Press the microwave start button | 4 |
| Restock pantry | 1: Fail to pick up the first item | 7 |
| | 2: Pick up the first item and place it in the pantry | 7 |
| | 3: Pick up the second item and place it in the pantry | 7 |
| Arrange vegetables | 1: Fail to pick up the first vegetable | 7 |
| | 2: Pick up the first vegetable and place it on the cutting board | 7 |
| | 3: Pick up the second vegetable and place it on the cutting board | 7 |
| Prepare coffee | 1: Fail to pick up the mug | 7 |
| | 2: Pick up the mug and place it in the coffee machine | 7 |
| | 3: Press the start button on the coffee machine | 7 |
| Pre-soak pan | 1: Fail to pick up the pan | 5 |
| | 2: Pick up the pan and place it in the sink | 5 |
| | 2: Pick up the sponge and place it in the pan | 5 |
| | 4: Turn the sink handle to turn on the sink | 5 |

## C  Details on evaluation tasks

### C.1  Frame-level task progress estimation

This task evaluates the model's ability to estimate the scalar progress of a robot toward task completion at every frame of a video, conditioned on a natural language task description. Ground-truth progress is calculated based on the simulator state at each moment of the video (Section 4.2). Given only task description and the video frames, the model must provide a numerical value reflecting task progress at each timestep. The evaluation metrics for this task include the L2 distance and the Pearson correlation coefficient between the predicted and ground-truth progress values across all frames. In addition, as a measure of reasoning quality that is independent of the ground-truth value calculations, we evaluate the correlation between the video frame count and the VLM-predicted task progress, which should decrease as videos deviate further from the known expert trajectory.

### C.2  Frame-level natural language reasoning

For this task, we evaluate the model's natural language text output as it reasons about what is happening at each frame of the video, such as "The robot gripper is moving closer to the mug" or "The robot is holding the carrot." The evaluation metrics include the frame-level reasoning error rate and success rate. The error rate is the percentage of frames where the generated description includes a statement that is verifiably false (i.e., directly contradicts information from the ground-truth simulator state at that moment). The success rate reflects the percentage of frames where the model states something that is verifiably true (i.e., is consistent with information from the ground-truth simulator state at that moment) and does not state something verifiably false. This task measures the model's capacity to precisely localize events in time, avoid hallucinations, and generate semantically faithful descriptions at a fine granularity.

To implement this for the results shown in Section 5, we treat the evaluation as a language model task, where we prompt the model with the frame description (generated by ROVER, GVL, or TemporalConcat) paired with the ground-truth information and ask if the description includes any statements that are verifiably true or false relative to the ground truth. We use `gpt-4.5-2025-02-27` as the language model for this evaluation. After reviewing 200 examples of the language model's evaluation by hand, we observed 97.5% alignment with human judgement. In Figure 14, we include examples below of frame-level reasoning from GVL versus ROVER where the model states something that, when compared to the ground-truth information, is verifiably false (red), verifiably true (green), or inconclusive (gray). In Figure 15, we include an example of where the LLM evaluation does not align with the human evaluation.

At each frame of the video, ROVER, along with baselines, generates a description of the frame before predicting task progress for that timestep. We cross-reference these descriptions with information about the ground-truth simulator state at each timestep to calculate the reasoning error rate and success rate (described in Appendix C.2). We treat this evaluation as a language model task, where we prompt the model with the generated description paired with the ground-truth information and ask if the description includes any statements that are verifiably true or false relative to the ground truth. We use `gpt-4.5-2025-02-27` as the language model for this evaluation (details in Appenedix C.2).

### C.3  Video question answering

The Video QA task tests a model's ability to reason over entire trajectories and answer questions about actions that may or may not have occurred. Each question is posed in the format: "Did the robot action? If so, when?" where action includes task-relevant operations like "contact the mug", "drop the potato", or "place the bowl in the microwave". Questions are automatically generated from a templated action grammar, grounded in the specific object and scene configuration of each video. The 27 robot manipulation tasks are grouped into 9 categories (Table 2). We ask similar questions within each task group based on the shared action types (Table 4). The evaluation metrics include:

- Accuracy: the percentage of questions for which the model correctly identifies whether the event occurred.
- Recall: the percentage of actual occurrences correctly identified by the model.
- Precision: the percentage of predicted occurrences that actually occurred in the video.

- Temporal distance: the difference in time between when something occurs during a video and when the VLM states that it occurred.

For the results shown in Section 5, we generate answers to each question for this task using the frame-level descriptions produced by ROVER, GVL, and TemporalConcat. For each question, we use a language model to generate an answer given all frame descriptions concatenated together. We again use gpt-4.5-2025-02-27 as the language model for this evaluation.

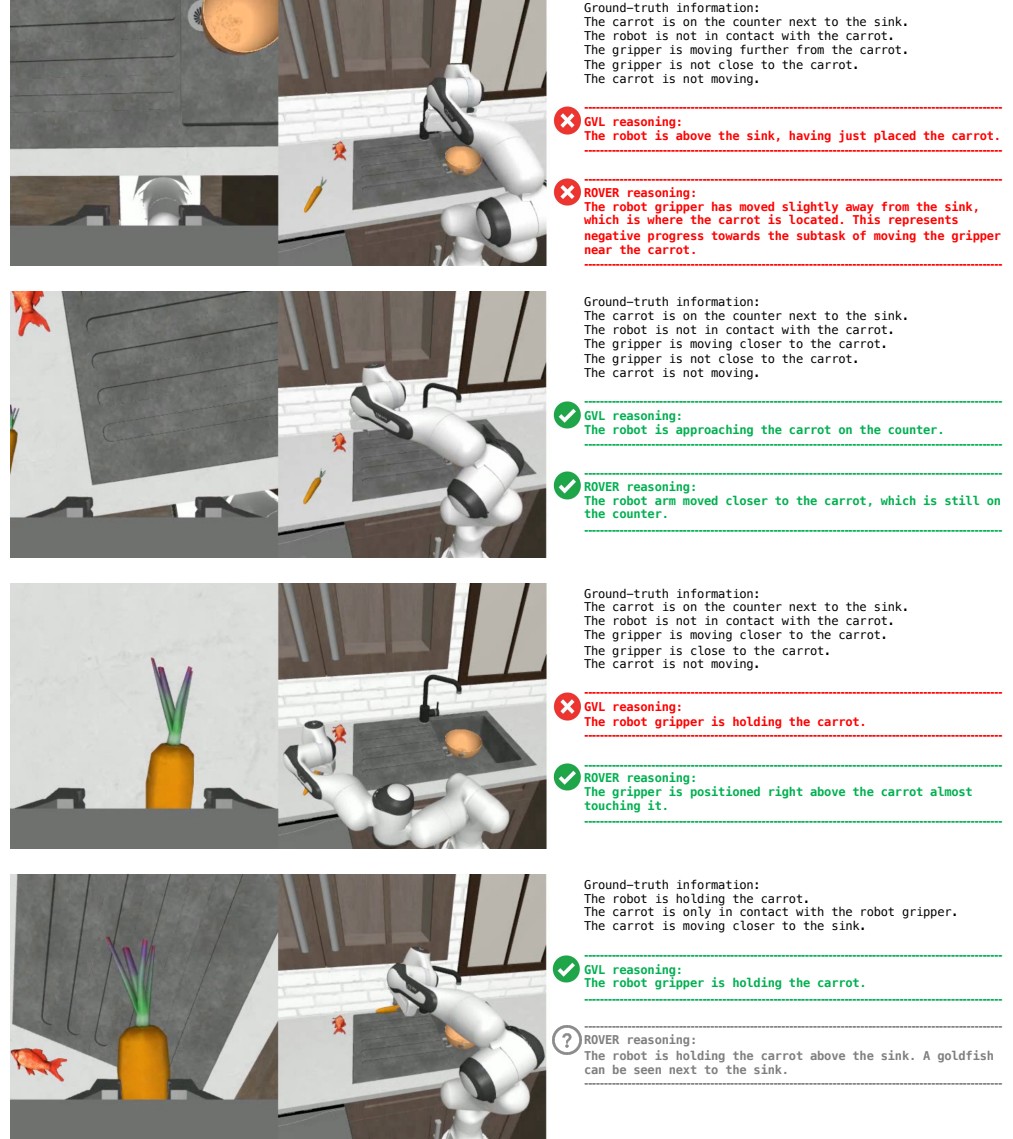

Figure 14: Examples of frame-level reasoning from GVL versus ROVER where the model states something that, when compared to the ground-truth information, is verifiably false (red), verifiably true (green), or inconclusive (gray).

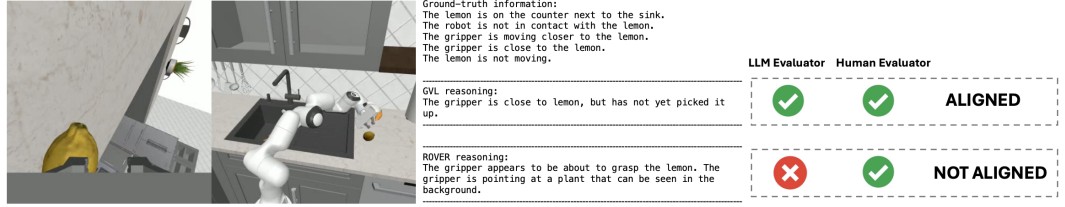

Figure 15: Examples of LLM evaluation compared to human evaluation of frame-level reasoning from GVL and ROVER.

Table 4: Questions for video QA task.

| Task group | Questions |
|---|---|
| Pick-and-place | Did the robot contact the {obj}? If so, when?
Did the robot pick up the {obj}? If so, when?
Did the robot drop the {obj}? If so, when?
Did the robot place the {obj} in {target_location}? If so, when? |
| Open/close doors/drawers | Did the robot contact the door/drawer handle? If so, when?
Did the robot grasp the door/drawer handle? If so, when?
Did the robot start opening/closing the door/drawer? If so, when?
Did the robot completely open/close the door/drawer? If so, when? |
| Turn on/off appliances | Did the robot press the start/stop button of the microwave/coffee machine? If so, when? |
| Toggle fixtures/knobs | Did the robot contact the sink spout/stove? If so, when?
Did the robot start turning the sink spout/sink handle/stove knob? If so, when?
Did the robot completely turn on/off the sink/stove? If so, when? |
| Microwave thawing | Did the robot open the microwave? If so, when?
Did the robot pick up the {obj}? If so, when?
Did the robot drop the {obj}? If so, when?
Did the robot place the {obj} in the microwave? If so, when? |
| Restock pantry | Did the robot pick up the {obj1}? If so, when?
Did the robot drop the {obj1}? If so, when?
Did the robot place the {obj1} in the cabinet? If so, when?
Did the robot pick up the {obj2}? If so, when?
Did the robot drop the {obj2}? If so, when?
Did the robot place the {obj2} in the cabinet? If so, when? |
| Arrange vegetables | Did the robot pick up the {vegetable1}? If so, when?
Did the robot drop the {vegetable1}? If so, when?
Did the robot place the {vegetable1} in the cabinet? If so, when?
Did the robot pick up the {vegetable2}? If so, when?
Did the robot drop the {vegetable2}? If so, when?
Did the robot place the {vegetable2} in the cabinet? If so, when? |
| Prepare coffee | Did the robot pick up the mug? If so, when?
Did the robot drop the mug? If so, when?
Did the robot place the mug in the coffee machine? If so, when?
Did the robot press the coffee machine start button? If so, when? |
| Pre-soak pan | Did the robot pick up the pan? If so, when?
Did the robot drop the pan? If so, when?
Did the robot place the pan in the sink? If so, when?
Did the robot pick up the sponge? If so, when?
Did the robot drop the sponge? If so, when?
Did the robot place the sponge in the pan? If so, when?
Did the robot contact the sink handle? If so, when?
Did the robot turn on the sink? If so, when? |

## C.4 Error types and examples

- Incorrect subtasks: When decomposing the task "put the mug in the coffee machine", the model specifies the subtask of "open the cabinet door" when the mug is on the counter.

- Redundant subtasks: The model first decomposes the task "put the steak in the microwave" into "pick up the steak from the counter" and "place the steak in the microwave", but then incorrectly decomposes the "place the steak in the microwave" subtask into "grasp the steak" and "move the steak to the microwave".

- Perception error: The model states that the gripper is holding an object when it is not.

- Reasoning error: Despite correctly identifying the gripper has dropped an object, the model states that task progress is increasing.

# D Prompting and implementation details

## D.1 GVL

Generative Value Learning (GVL) [31] is the existing state-of-the-art in-context learning approach for VLM reasoning over video input for embodied tasks. The method is motivated by the observation in [31] that, when presented a choronological sequence of video frames, VLMs often ignore the trajectory quality and instead hallucinate monotonically increasing task progress. To disrupt this temporal bias, GVL randomly shuffles the input frames. In doing so, GVL forces the model to pay attention to what is happening to each frame, improving frame-level task progress prediction. We implement GVL using the following prompt from [31].

```
************************** SYSTEM PROMPT
You are an expert roboticist tasked to predict task completion
percentages for frames of a robot for the task of {task_description}.
Note that the robot has an unknown level of expertise with the task
and may perform actions that lead it signicantly further from
accomplishing the task. Note that these frames are in random order,
so please pay attention to the individual frames when reasoning
about task completion percentage. If the progress at the current
frame is less than the progress of the initial robot scene, then the
task completion percentages should be negative.

For the task of {task_description}, output the task completion
percentage for the following frames that are presented in random
order. For each frame, format your response as follows:
Frame description: {}
Task completion percentage: {}%

************************** TASK PROMPT
Initial robot scene: [IMG]
Frame description: {first_frame_description}
Task completion percentage: 0%

Frame 1: [IMG]
...
...
...
Frame n: [IMG]
```

## D.2 ROVER

Similar to GVL, we implement ROVER using an in-context learning approach, with the same prompting, shown below, used for all backbone VLMs. As discussed in Section 3, for each frame, the VLM describes the frame and either specifies a new subtask that needs to be completed or predicts progress for the existing subtask. If a new subtask is specified, then a new line of reasoning is created (using the same prompting shown below) with the updated {task_description} for the new subtask. To specify a new subtask, the VLM generates the text tokens "The robot needs to: {new_subtask}" as depicted in Figure 1. If a new subtask is not specified, the VLM proceeds with the frame-by-frame progress prediction for the existing subtask. To move to the next frame, the VLM generates the text tokens "[next-frame]" as shown in the prompting below. As described in Section 3, we implement a sliding window when appending the next frame, $o_{next}$, to the existing context, $Y$. When appending a new frame, $Y = \text{Window}(Y + o_{next})$, the window extracts three frames: the first frame of the current line of reasoning and the most recent previous frame (each followed by their previously generated natural language descriptions), along with the newly added next frame. The Window function formats these three frames as shown in the prompt below.

```
************************** SYSTEM PROMPT
You are an expert roboticist tasked to predict subtask completion
percentages for frames of a robot for the subtask of {task_description}.

The frames are shown in temporal order. Frame 0 represents the beginning
of the subtask. Note that the robot has an unknown level of expertise with
the subtask and may perform actions that lead it signicantly further from
accomplishing the subtask. Therefore, please pay attention to the
individual frames when reasoning about subtask completion percentage. If
the progress at the current frame is less than the progress of the initial
robot scene, then the completion percentages should be negative.

If the given subtask can be decomposed into multiple subtasks, please
specify a new subtask. Some examples of decomposition are provided below.
Subtask: 'pick the cheese from the counter and place it in the cabinet'
New Subtasks: ['grasp the cheese', 'place the cheese in the cabinet']
...
...

If you decompose the subtask further or progress to the next subtask,
format your response as follows:
Frame description: {}
The robot needs to: {}

If you do not decompose the subtask further or progress to the next
subtask, format your response as follows:
Frame description: {}
Subtask completion percentage: {}%
[next-frame]

************************** TASK PROMPT
Initial robot scene: [IMG]
Frame description: {first_frame_description}
Subtask completion percentage: 0%

###### If VLM has progressed beyond second frame of existing subtask
Most recent previous frame: [IMG]
Frame description: {prev_frame_description}
Subtask completion percentage: {prev_subtask_progress}%
######

Current frame: [IMG]
```

# E Simulation environment and dataset of expert demonstrations

We leverage RoboCasa [39] as the simulation environment, which includes a large dataset of expert demonstrations, collected via human teleoperation, for household tasks in its initial release. RoboCasa is a large-scale simulation framework centered around home environments for training generalist robots. RoboCasa builds upon RoboSuite [66], a framework based in MuJoCo with a focus on physical realism, high speed, and modular design. RoboCasa contains 120 kitchen scenes and over 2,500 3D assets spanning 153 unique object categories, created with the aid of generative AI tools [39] (depicted in Figures 16 and 17 from [39]). RoboCasa includes a dataset of demonstrations collected through human teleoperation. A team of four human operators collected 50 high-quality demonstrations for each atomic task, along with 5 composite tasks, using a 3D SpaceMouse [65, 66].

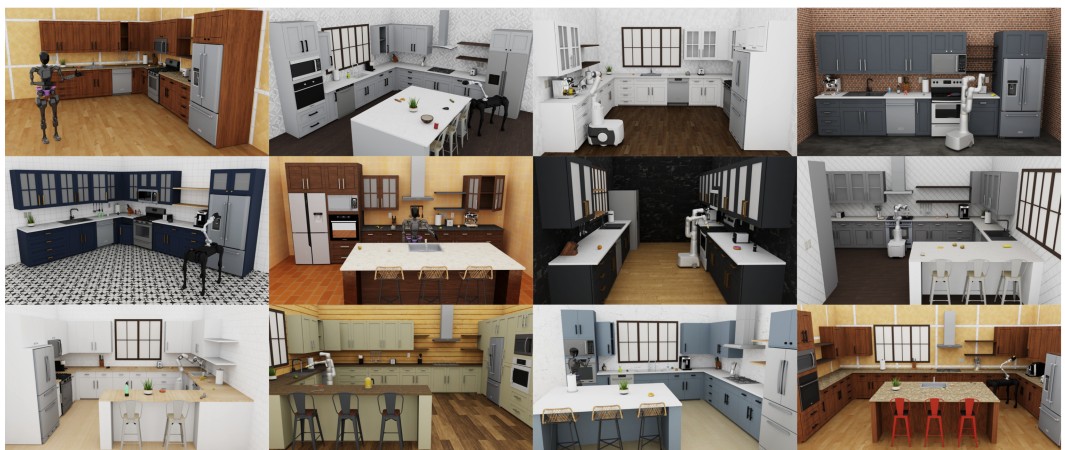

Figure 16: Figure from [39] showing RoboCasa kitchen scenes.

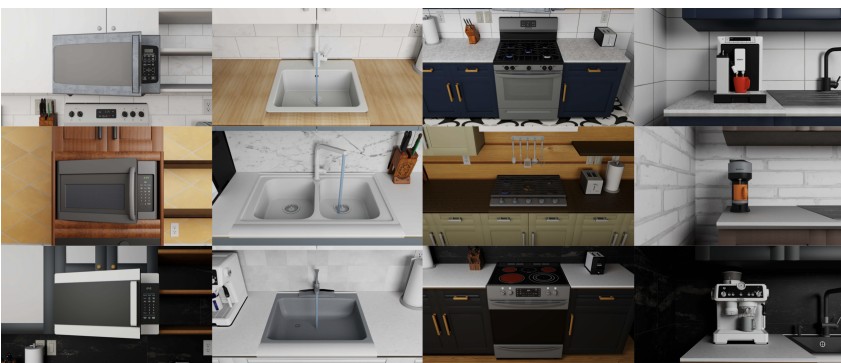

Figure 17: Figure from [39] showing RoboCasa interactable appliances.

# F Analysis of inference cost

We measure two key indicators of inference cost across tasks with different video lengths: (1) the total number of tokens processed per video and (2) the number of VLM calls. The tables below show that ROVER consistently reduces token count compared to baseline methods, even though it requires a slightly higher number of model calls due to its recursive structure. Importantly, the increase in inference cost with respect to video length is approximately linear, consistent with our design goal. For instance, doubling the video length from 30 to 60 frames results in approximately a doubling of token usage for ROVER (from 47k to 95k tokens), supporting the linear scaling claim. Due to ROVER scaling linearly with video length (with baselines scaling quadratically), the improvements in inference cost we observe with ROVER increase as video length increases. For videos with 30 frames (the single-stage atomic tasks), ROVER shows about a 3x reduction in the number of tokens per video. For videos with 60 frames (the multi-stage composite tasks), ROVER shows more than a 5x reduction in the number of tokens per video.

Table 5: Inference cost of each method for videos with 30 frames.

| Method | Avg number of tokens per video | Avg number of VLM calls per video |
|---|---|---|
| ROVER | 47,650 | 36 |
| GVL | 140,307 | 30 |
| TemporalConcat | 134,855 | 30 |

Table 6: Inference cost of each method for videos with 60 frames.

| Method | Avg number of tokens per video | Avg number of VLM calls per video |
|---|---|---|
| ROVER | 95,404 | 68 |
| GVL | 539,102 | 60 |
| TemporalConcat | 527,572 | 60 |

# G Extended baselines

The results below include two recent video-specific baselines: VideoLLaMA-3 and Video-LLaVA. These models are designed specifically for video-language tasks and incorporate temporal modeling and memory mechanisms that target long-sequence video comprehension. We also include the baseline VideoGemini that concatenates all frames of the video to provide as input to the model.

Table 7: Correlation with ground-truth values for frame-level progress prediction with extended baselines.

| Method | Real-world videos | Simulation videos |
|---|---|---|
| ROVER | 0.62 | 0.56 |
| GVL | 0.31 | 0.35 |
| TemporalConcat | 0.43 | 0.17 |
| LocalConcat | 0.46 | 0.18 |
| LIV | 0.12 | 0.09 |
| VideoGemini | 0.46 | 0.29 |
| VideoLlama3 | 0.34 | 0.21 |
| VideoLLaVA | 0.19 | 0.14 |

Table 8: Reasoning error rate for frame-level natural language reasoning with extended baselines.

| Method | Error rate |
|---|---|
| ROVER | 45% |
| GVL | 59% |
| TemporalConcat | 67% |
| LocalConcat | 62% |
| VideoGemini | 62% |
| VideoLlama3 | 71% |
| VideoLLaVA | 76% |

Table 9: Accuracy for video QA with extended baselines.

| Method | Accuracy |
|---|---|
| ROVER | 77% |
| GVL | 51% |
| TemporalConcat | 28% |
| LocalConcat | 25% |
| VideoGemini | 68% |
| VideoLlama3 | 62% |
| VideoLLaVA | 58% |

# H  Stratifying results by state type and context length

GVL and TemporalConcat attempt to reason over the full video by concatenating all frames together. Figure 18 shows the correlation GVL and TemporalConcat achieve with ground-truth value estimates when reasoning over 0-10 frames, 10-20 frames, and 20-30 frames. In addition, the figure further stratifies results based on the state type at the current timestep (expert, non-expert, or interpolating between expert and non-expert) of the trajectory. These are the same results presented in Section 5.2, but now stratified across these two dimensions (instead of stratifying across trajectory level, as done in Figure 3). We see that the performance of GVL and TemporalConcat degrades when encountering non-expert moments of the trajectory, especially when the number of frames included in the context extends beyond 10. We observe a similar trend below when stratifying the error rates with the frame-level natural language reasoning across context length and state type (Figure 19). ROVER limits the maximum number of frames the model must reason over to three frames using the sliding context window (described in Section 3). In doing so, ROVER avoids these errors observed in long-context reasoning.

## H.1 Frame-level task progress estimation

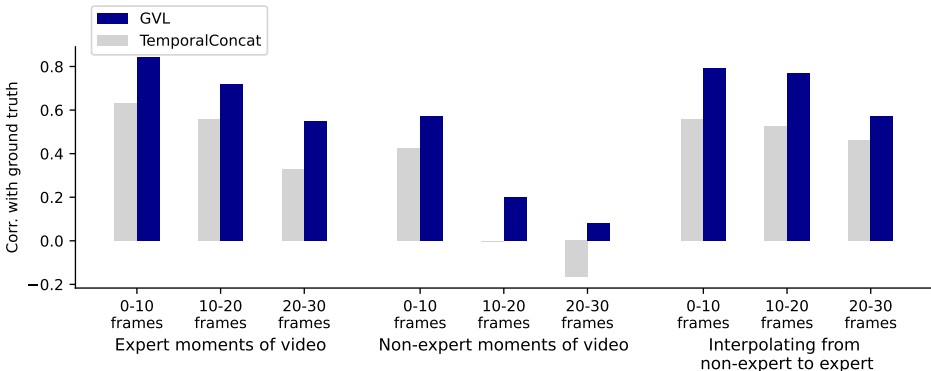

Figure 18: Average correlation GVL and TemporalConcat achieve with ground-truth value estimates in the frame-level progress prediction task when reasoning over frame sequences of different lengths. .

## H.2 Frame-level natural language reasoning

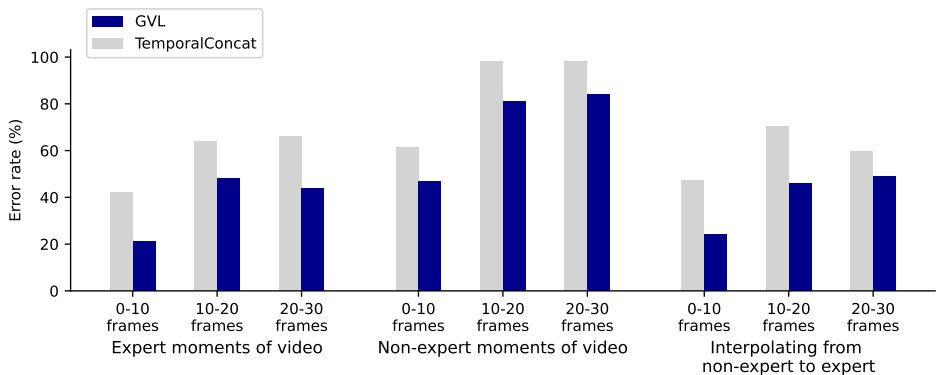

Figure 19: Average error rate of GVL and TemporalConcat with the frame-level natural language reasoning task when reasoning over frame sequences of different lengths.

# H  Robustness to video length and frame rate

The total frame count of a video is determined by two factors: 1) the video length: the total amount of time that the video represents and 2) the frame rate: the number of image frames used to represent each unit of time. These two factors can vary widely for a given video at test time. Therefore, for a method to achieve reliable performance in the wild, it should be robust to both factors.

**ROVER scales more readily to longer videos and higher frame rates.** We see that ROVER, by leveraging recursive reasoning and using a maximum context of three frames, is not only more efficient, but is also less sensitive to video length and frame rate of the given video relative to GVL and TemporalConcat. We first observe this in the results stratified across task group (Figures 3 and 5 from the main results in Section 5), where we see the improved performance of ROVER for the composite tasks (bottom row of each figure), which include much longer videos than the atomic tasks (top row of each figure). We also observe ROVER's robustness to frame rate when varying the total number of frames used to represent each video between 30 and 240 (i.e., increasing the effective frame rate by a factor of 8) for the 10 pick-and-place tasks for the progress prediction task (Figure 20) and frame-level natural language reasoning (Figure 21). These figures show that the performance of GVL and TemporalConcat significantly degrades when given the same pick-and-place videos, but with effective frame rate increased.

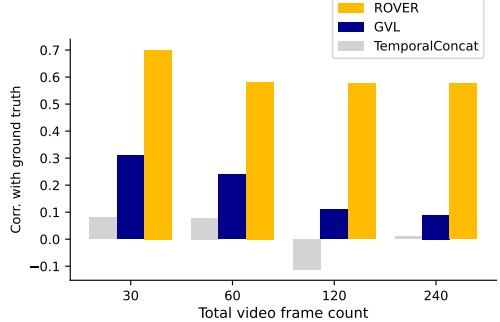

Figure 20: Average correlation each method achieves with ground-truth value estimates in the frame-level progress prediction task when increasing the total frame count from 30 to 240 (by increasing the frame rate) for the 10 pick-and-place tasks.

Figure 21: Average error rate of each method with the frame-level natural language reasoning task when increasing the total frame count from 30 to 240 (by increasing the frame rate) for the 10 pick-and-place tasks.

# I  Robustness to camera view

The main results presented in Section 5 use the wrist view (eye in hand view), as all methods were observed to perform best on the manipulation tasks using this view. The figures below show the results of the progress prediction task (Figure 22) and frame-level natural language reasoning task (Figure 23) when using different camera views. The left view and right view are fixed third-person views with the camera positioned to the left and right, respectively, of the robot at a distance sufficent to show the full robot arm within the camera frame. Although all methods decline in performance with these fixed external views (likely due to occlusions of the object or gripper at different moments during the trajectory), ROVER continues to outperform baselines across all camera views.

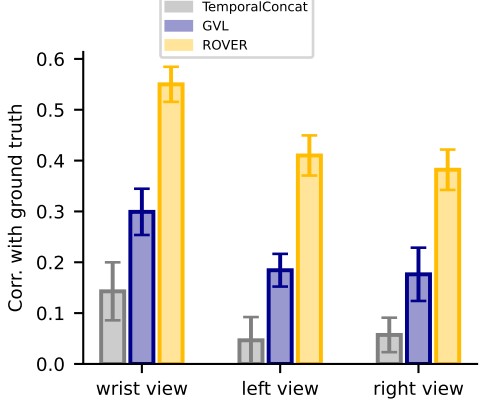 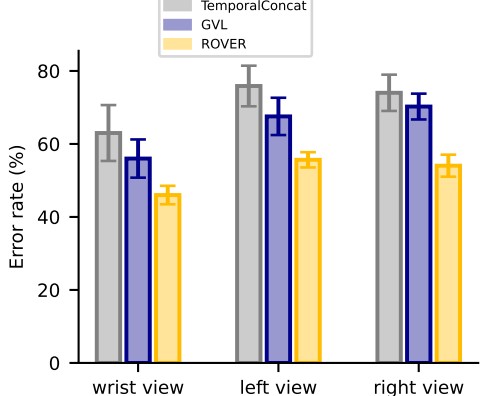

Figure 22: Mean and standard error of correlation between ground-truth value estimates and progress values predicted by each method stratified across different camera views.

Figure 23: Mean and standard error of error rate in frame-level natural language reasoning of each method stratified across different camera views.

# J  Testing across different backbone models

To ensure consistency across benchmarks and with prior work, We use `gemini-1.5-pro` for Gemini-1.5-Pro and `gemini-2.5-pro-preview` for Gemini-2.5-Pro-Preview from Google's Gemini API. We use `gpt-4o` from the OpenAI API for GPT-4o. We use `Qwen2.5-VL-32B-Instruct` for Qwen2.5-VL-32B available on Huggingface. For the open source model, all experiments were performed on A6000 GPUs. Overall, we see that ROVER consistently outperforms baselines across all backbone VLMs.

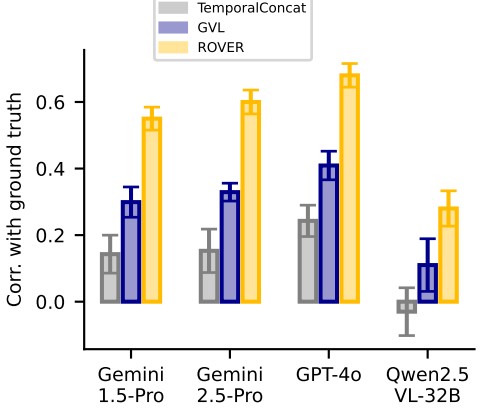

Figure 24: Mean and standard error of correlation between ground-truth value estimates and progress values predicted by each method stratified across different backbone VLMs.

Figure 25: Mean and standard error of distance between ground-truth value estimates and progress values predicted by each method stratified across different backbone VLMs.

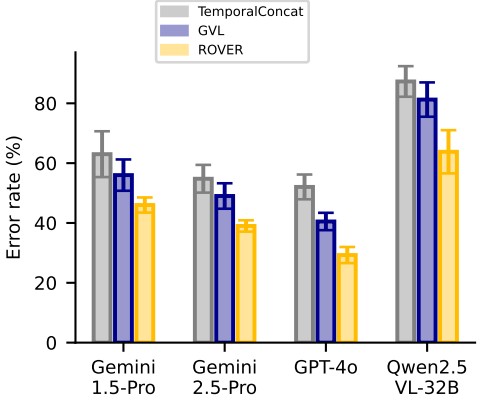

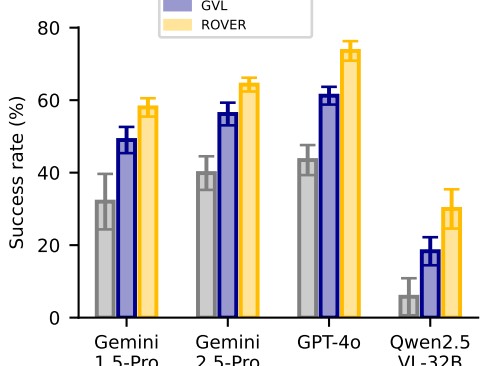

Figure 26: Mean and standard error of error rate in frame-level natural language reasoning of each method stratified across different backbone VLMs.

Figure 27: Mean and standard error of success rate in frame-level natural language reasoning of each method stratified across different backbone VLMs.

## K Task progress prediction with real-world OpenX Embodiment datasets

We perform the same task progress prediction analysis conducted in [31] using 1,000 videos from 50 real-world OpenX Embodiment datasets (20 videos randomly selected from each dataset) [41]. The OXE dataset includes an aggregation of trajectory data from 50 standalone robot datasets that consists of diverse tasks, robots, and camera viewpoints. The OXE videos include 20 robot embodiments, over 300 different task specifications, and highly diverse (often non-ideal) camera viewpoints.

The tables below show the average correlation each method achieves with the ground-truth value estimates across the OXE datasets when using Gemini-1.5-Pro (Table 10) and GPT-4o (Table 11) as the backbone VLM. Since there are no true value estimates for these real-world videos, we follow [31] and use frame number as the ground-truth value estimate. For known high-quality datasets collected from human teleoperators with fixed camera placements, such as Bridge [13], RT-1 [7], and Dobb-E [47], ROVER achieves a significantly higher correlation with frame number than GVL (highlighted in yellow in the tables below). Both ROVER and GVL show low correlation with frame number for datasets with videos that show lower quality trajectories produced by autonomous data collection via scripted motions or motor babbling, such as QT-OPT [23] and RoboNet [10] (highlighted in green in the tables below).

Table 10: Correlation of each method with ground-truth progress estimates (based on frame number) on OpenX Embodiment datasets with Gemini-1.5-Pro as the backbone VLM.

| Dataset | GVL | ROVER |
|---|---|---|
| `utokyo_xarm_pick_and_place_converted_externally_to_rlds` | 0.753 | 0.898 |
| `utokyo_xarm_bimanual_converted_externally_to_rlds` | 0.822 | 0.890 |
| `nyu_door_opening_surprising_effectiveness` | 0.668 | 0.879 |
| `berkeley_autolab_ur5` | 0.761 | 0.874 |
| `utokyo_pr2_tabletop_manipulation_converted_externally_to_-rlds` | 0.619 | 0.870 |
| `maniskill_dataset_converted_externally_to_rlds` | 0.740 | 0.866 |
| `utokyo_pr2_opening_fridge_converted_externally_to_rlds` | 0.824 | 0.856 |
| `fractal20220817_data` | 0.762 | 0.817 |
| `iamlab_cmu_pickup_insert_converted_externally_to_rlds` | 0.513 | 0.805 |
| `toto` | 0.564 | 0.780 |
| `ucsd_kitchen_dataset_converted_externally_to_rlds` | 0.529 | 0.768 |
| `utaustin_mutex` | 0.576 | 0.767 |
| `asu_table_top_converted_externally_to_rlds` | 0.477 | 0.762 |
| `austin_sirius_dataset_converted_externally_to_rlds` | 0.539 | 0.755 |
| `dobbe` | 0.487 | 0.754 |
| `berkeley_cable_routing` | 0.488 | 0.752 |
| `berkeley_rpt_converted_externally_to_rlds` | 0.495 | 0.748 |
| `viola` | 0.415 | 0.747 |
| `fmb` | 0.555 | 0.739 |
| `austin_buds_dataset_converted_externally_to_rlds` | 0.350 | 0.727 |
| `usc_cloth_sim_converted_externally_to_rlds` | 0.457 | 0.722 |
| `bridge` | 0.457 | 0.717 |
| `jaco_play` | 0.414 | 0.710 |
| `stanford_hydra_dataset_converted_externally_to_rlds` | 0.351 | 0.701 |
| `bc_z` | 0.397 | 0.696 |
| `berkeley_mvp_converted_externally_to_rlds` | 0.360 | 0.680 |
| `cmu_stretch` | 0.276 | 0.677 |
| `tokyo_u_lsmo_converted_externally_to_rlds` | 0.362 | 0.639 |
| `berkeley_fanuc_manipulation` | 0.252 | 0.632 |
| `roboturk` | 0.314 | 0.631 |
| `ucsd_pick_and_place_dataset_converted_externally_to_rlds` | 0.281 | 0.615 |
| `dlr_edan_shared_control_converted_externally_to_rlds` | 0.119 | 0.598 |
| `dlr_sara_pour_converted_externally_to_rlds` | 0.115 | 0.564 |
| `droid` | -0.002 | 0.546 |
| `taco_play` | 0.083 | 0.539 |
| `stanford_robocook_converted_externally_to_rlds` | -0.010 | 0.513 |
| `imperialcollege_sawyer_wrist_cam` | -0.038 | 0.432 |
| `kaist_nonprehensile_converted_externally_to_rlds` | -0.021 | 0.425 |
| `austin_sailor_dataset_converted_externally_to_rlds` | -0.072 | 0.371 |
| `kuka` | 0.304 | 0.316 |
| `cmu_play_fusion` | -0.096 | 0.304 |
| `stanford_kuka_multimodal_dataset_converted_externally_to_-rlds` | -0.110 | 0.295 |
| `nyu_franka_play_dataset_converted_externally_to_rlds` | -0.139 | 0.287 |
| `stanford_mask_vit_converted_externally_to_rlds` | -0.131 | 0.270 |
| `uiuc_d3field` | -0.212 | 0.169 |
| `robo_net` | -0.240 | 0.096 |
| `columbia_cairlab_pusht_real` | -0.229 | 0.095 |
| `dlr_sara_grid_clamp_converted_externally_to_rlds` | -0.301 | -0.024 |
| `cmu_franka_exploration_dataset_converted_externally_to_-rlds` | -0.230 | -0.078 |

Table 11: Correlation of each method with ground-truth progress estimates (based on frame number) on OpenX Embodiment datasets with GPT-4o as the backbone VLM.

| Dataset | GVL | ROVER |
|---|---|---|
| utokyo_pr2_opening_fridge_converted_externally_to_rlds | 0.907 | 0.969 |
| nyu_door_opening_surprising_effectiveness | 0.891 | 0.928 |
| berkeley_autolab_ur5 | 0.749 | 0.894 |
| utaustin_mutex | 0.837 | 0.893 |
| berkeley_mvp_converted_externally_to_rlds | 0.835 | 0.889 |
| fractal20220817_data | 0.788 | 0.879 |
| utokyo_xarm_bimanual_converted_externally_to_rlds | 0.714 | 0.874 |
| utokyo_pr2_tabletop_manipulation_converted_externally_to_rlds | 0.722 | 0.867 |
| austin_sirius_dataset_converted_externally_to_rlds | 0.717 | 0.867 |
| toto | 0.717 | 0.851 |
| dlr_edan_shared_control_converted_externally_to_rlds | 0.696 | 0.840 |
| dobbe | 0.568 | 0.829 |
| iamlab_cmu_pickup_insert_converted_externally_to_rlds | 0.554 | 0.824 |
| utokyo_xarm_pick_and_place_converted_externally_to_rlds | 0.726 | 0.822 |
| ucsd_kitchen_dataset_converted_externally_to_rlds | 0.629 | 0.814 |
| berkeley_fanuc_manipulation | 0.603 | 0.807 |
| viola | 0.503 | 0.804 |
| asu_table_top_converted_externally_to_rlds | 0.505 | 0.803 |
| bridge | 0.661 | 0.799 |
| maniskill_dataset_converted_externally_to_rlds | 0.527 | 0.787 |
| jaco_play | 0.570 | 0.786 |
| berkeley_rpt_converted_externally_to_rlds | 0.616 | 0.781 |
| roboturk | 0.586 | 0.769 |
| usc_cloth_sim_converted_externally_to_rlds | 0.466 | 0.765 |
| uiuc_d3field | 0.524 | 0.730 |
| austin_buds_dataset_converted_externally_to_rlds | 0.458 | 0.728 |
| kaist_nonprehensile_converted_externally_to_rlds | 0.483 | 0.704 |
| austin_sailor_dataset_converted_externally_to_rlds | 0.353 | 0.681 |
| berkeley_cable_routing | 0.278 | 0.676 |
| tokyo_u_lsmo_converted_externally_to_rlds | 0.394 | 0.674 |
| ucsd_pick_and_place_dataset_converted_externally_to_rlds | 0.245 | 0.663 |
| cmu_play_fusion | 0.374 | 0.661 |
| dlr_sara_pour_converted_externally_to_rlds | 0.244 | 0.615 |
| imperialcollege_sawyer_wrist_cam | 0.219 | 0.602 |
| stanford_robocook_converted_externally_to_rlds | 0.206 | 0.598 |
| stanford_hydra_dataset_converted_externally_to_rlds | 0.178 | 0.598 |
| cmu_stretch | 0.140 | 0.598 |
| bc_z | 0.158 | 0.591 |
| nyu_franka_play_dataset_converted_externally_to_rlds | 0.182 | 0.567 |
| robo_net | 0.367 | 0.438 |
| stanford_kuka_multimodal_dataset_converted_externally_to_rlds | -0.042 | 0.434 |
| stanford_mask_vit_converted_externally_to_rlds | -0.042 | 0.433 |
| columbia_cairlab_pusht_real | -0.048 | 0.416 |
| eth_agent_affordances | -0.088 | 0.416 |
| cmu_franka_exploration_dataset_converted_externally_to_rlds | -0.064 | 0.356 |
| taco_play | -0.094 | 0.299 |
| kuka | 0.150 | 0.170 |
| dlr_sara_grid_clamp_converted_externally_to_rlds | -0.301 | -0.047 |

# L Ablations

As described in Section 5.5, we test two ablations of ROVER: one variant ablates only the sliding window while the other ablates only the recursive reasoning. We observe that baseline methods such as TemporalConcat often hallucinate steady, monotonically increasing task progress values, even when the video includes significant stalls or regressions. GVL addresses this problem via random frame shuffling, which disrupts the strong temporal priors and forces the model to assess each frame more carefully [31]. We observe that the sliding window, even when ablating the recursive decomposition, similarly reduces hallucination by retaining temporal context while avoiding long-context sequential processing. Our ablations show that the window-only variant of ROVER significantly outperforms TemporalConcat and matches or exceeds GVL's performance on short-horizon tasks, such as turning on/off appliances or toggling fixtures/knobs. We see that adding the recursive decomposition further improves performance for more complex tasks consisting of numerous subtasks.

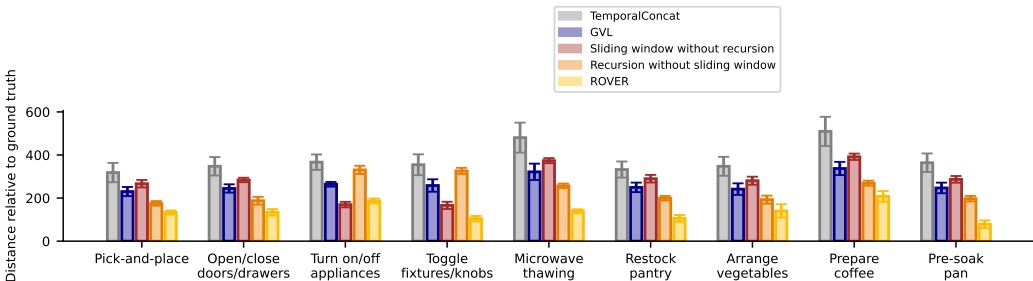

Figure 28: Mean and standard error of distance between ground-truth value estimates and progress values predicted by baselines and ablated versions of ROVER stratified by task group.

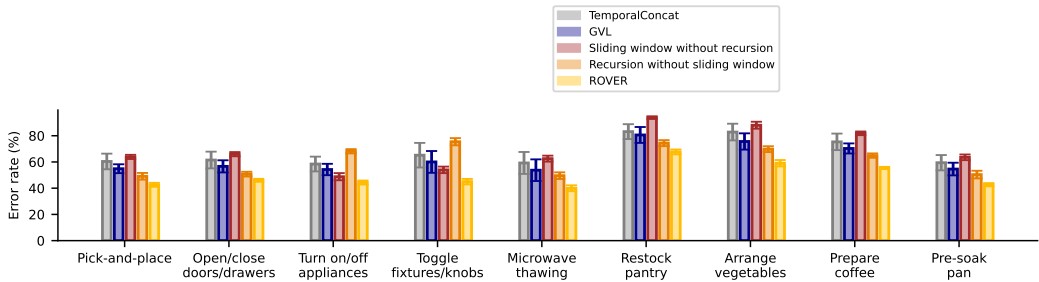

Figure 29: Mean and standard error of error rate in frame-level natural language reasoning of baselines and ablated versions of ROVER stratified by task group.

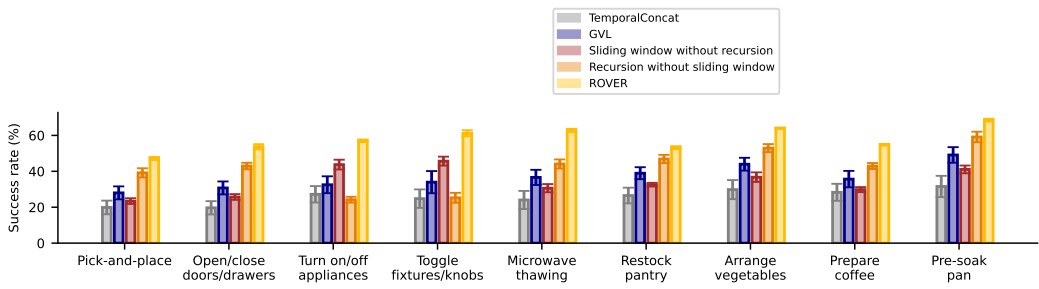

Figure 30: Mean and standard error of success rate in frame-level natural language reasoning of baselines and ablated versions of ROVER stratified by task group.

The sliding window of ROVER includes 3 frames: the first frame of the current line of reasoning, the most recent previous frame, and the frame for the current timestep of the video. In the table below, we show the performance of ROVER on the frame-level progress prediction task when changing the sliding window size to 6, 9, and 12 frames. We see that, as the number of frames included in the sliding window increases, the performance of ROVER drops.

To evaluate the cause of performance drop when increasing the sliding window size, we apply the same error analysis that we describe above to the modified versions of ROVER that have increasing sliding window size. Based on the results of this analysis shown in the table below, the drop in performance with increasing window size appears to be driven by increases in perception errors, which also comprise the primary error type observed with GVL. Perception errors occur when the model hallucinates basic facts about what is happening in the video. We see that, in the original version of ROVER, only 5% of videos include a perception error, but in the version of ROVER with a window size of 12, this rises to 19% of videos.

This supports our findings that the VLM is prone to perception errors when attempting to reason over many frames of a video. ROVER (with a sliding window size of 3) mitigates these perception errors by shortening the number of frames the model must reason over at each timestep of the video. When the sliding window size is increased, perception errors increase. This is also consistent with our finding from the initial error analysis that 83% of perception errors from GVL occur when there are over 10 frames of the video included in the context.

We see that these modified versions of ROVER still do not produce as many perception errors as GVL (which makes a perception error on over 30% of videos, as shown in Table R14). This is likely due to the fact that, even with a sliding window size of 12, ROVER still shortens the model's context at each moment of reasoning by decomposing the video into subtask-specific lines of reasoning (which are generally less than 12 frames in length, even without implementing a sliding window).

Overall, we find that ROVER introduces the possibility of decomposition errors, but dramatically reduces the total error rate by mitigating the model's tendencies toward hallucination (perception errors).

Table 12: Correlation with ground-truth values for frame-level progress prediction (with varying sliding context window size).

| Method | Corr. with Ground Truth |
|---|---|
| ROVER (original) | 0.56 |
| ROVER (window of 6) | 0.52 |
| ROVER (window of 9) | 0.50 |
| ROVER (window of 12) | 0.44 |

Table 13: Error rates (%) observed with increasing sliding window size for ROVER.

| Error type | window of 3 (original) | window of 6 | window of 9 | window of 12 |
|---|---|---|---|---|
| Incorrect subtasks | 7% | 6% | 5% | 7% |
| Redundant subtasks | 4% | 3% | 5% | 4% |
| Perception error | 5% | 10% | 12% | 19% |
| Reasoning error | 4% | 4% | 5% | 6% |
| Other | 3% | 2% | 4% | 4% |
| Total | 23% | 25% | 31% | 40% |

# M   Additional details on object-centric subtasks

When generating non-expert trajectories (described in Section 4.1) and computing ground-truth value estimates (described in Section 4.2), we follow the work in MimicGen [36] in defining object-centric subtasks for each task trajectory. Specifically, if we let $E = \{e_1, e_2, ...\}$ be the set of objects in a task $\mathcal{M}$, we assume that each expert demonstration, $\tau^E \in \mathcal{D}_{src}$, can be decomposed into object-centric segments $\tau^E = \{\tau_i^E(e_{\tau_i})\}_{i=1}^M$, where the manipulation in each subtask trajectory is relative to a single object's coordinate frame ($e_{\tau_i} \in E$). This sequence of object-centric subtasks is generally straight forward for a human to identify given a task [36]. For example, for the pick-and-place task of "pick the cup from the counter and place it in the sink", the subtasks include first picking up the cup, then placing it in the sink. This task can be broken down into two object-centric subtasks: a cup-grasping subtask (motion is relative to cup) and a cup-placement subtask (motion is relative to the sink basin). Similar to MimicGen [36], we assume access to metrics that allow the end of each subtask to be detected. In the previous example, this would correspond to detecting 1) when the cup has been lifted from the counter and 2) when the cup has been placed within the sink basin.

We generate the non-expert trajectories (by inserting random perturbations within the expert demonstrations as described in Section 4.1) and compute the value estimates (based on the distance measures described in Section 4.2) for each of these object-centric subtasks separately. To create the full non-expert trajectory, $\tau^N = \{\tau_i^N(e_{\tau_i^N})\}_{i=1}^M$, we simply sequence together the subtask trajectories. To form the value estimates for the full trajectory, we sequence together the value estimates corresponding to each subtask, adding the value estimates of each subtask to the final value of the previous subtask and, when complete, rescaling the list of values to the range $[0, 1]$.

# O   Full related work

**Decomposition-based reasoning.**   ROVER enables more precise and efficient reasoning over video for long-horizon embodied tasks by segmenting the video input into subtask-specific segments through recursive decomposition. Prior work has explored decomposition in various forms, including neural modular architectures [2, 17], multi-hop QA [50], and multi-step reasoning for program synthesis or mathematical problems [37, 16, 28]. More recently, LLMs have been used to decompose problems via in-context learning or fine-tuning for complex tasks such as code generation and planning [55, 24, 64]. Adaptive decomposition methods have also emerged, including feedback-driven re-planning [42, 44] and multi-agent coordination [59]. Unlike these approaches, ROVER performs recursive decomposition over video input with VLMs. Further, we show that ROVER can be used to improve reasoning about physical interactions in embodied settings.

**Goal-conditioned value functions and reward modeling.**   ROVER demonstrates an emerging use of VLMs as per-frame value estimators for embodied video tasks, leveraging their semantic understanding and long-context capabilities. Prior work developed models using robot [46] or human videos with discriminators [8], contrastive learning [4], or offline reinforcement learning [34, 32, 5]. With the advent of foundation models, researchers have begun incorporating language and vision models into downstream robotic applications including planning [1, 48, 12], imitation learning [6, 49], and symbolic reasoning [30, 20, 51].

[27] and [35] leverage language models to assign reward values to reinforcement learning agents. [26], [58], and again [27] utilize these models to deliver preference-based feedback. Works such as [33], [62], and [60] go further by having LLMs generate code. Notably, all these approaches rely solely on the language understanding capabilities of foundation models.

Prior methods leveraging VLMs for reward prediction over video input primarily relied on two extremes: reasoning over a small set of frames (e.g., comparing two individual frames) [57, 25, 54, 63] or attempting to reason over the full video by concatenating all frames into a single context [31]. The former sacrifices global context, while the latter is inefficient (inference cost scales quadratically with video length) and overwhelms the model with visual information that is often redundant or irrelevant. ROVER addresses these limitations by leveraging recursive decomposition to enable high-precision frame-by-frame reasoning (including reward or task progress prediction), without sacrificing efficiency or global context.

