# OpenReview forum: "ROVER: Recursive Reasoning Over Videos with Vision-Language Models for Embodied Tasks"
_NeurIPS.cc/2025/Conference — NeurIPS 2025 poster_

### Official Review · Reviewer_JuQn · 2025-06-30

**Clarity:** 4
**Significance:** 4
**Originality:** 3
**Rating:** 5
**Confidence:** 4

**Summary:**

The paper proposes an in-context learning approach paired with a recursive algorithm to improve VLM's ability at video understanding for embodied videos. Prior works have to balance global reasoning (on the entire video) and localized reasoning (on a small set of frames). In contrast, the proposed approach, ROVER, identifies subtasks, which recursively spawn another instance of ROVER to reason about the subtask. ROVER also maintains a sliding window that includes context from the previous subtask and the first frame of the current subtask. The paper tests ROVER on a curated dataset of robot videos with various levels of expertise (where worse trajectories are achieved via random perturbation) and videos from the Open-X dataset, showing higher accuracy, especially on suboptimal videos, over other baselines, which tend to hallucinate successes.

**Questions:**

* What are some failure cases for incorrect/unncessary subtasks generated by ROVER?
* In Figure 5, why does the ROVER somtimes exhibit an increasing trend in error rate even though the level of expertise in a video increases (e.g., for pick-and-place, and prepare-coffee)?
* What is the size of the sliding window? Is it just 3 frames? Or is it the most recent previous frame, the first frame of the current line of reasoning, plus N number of frames? How does the sliding window size affect ROVER's performance?

Some additional questions are sprinkled throughout the weaknesses sections.

**Ethical Concerns:**

["NO or VERY MINOR ethics concerns only"]

**Final Justification:**

I appreciate the author's follow-up responses, and they have addressed most of my concerns and questions. I will increase my score accordingly. I expect the detailed analysis (e.g., failure cases, inference cost) and additional baselines to be included in the final version of the paper.

**Limitations:**

Yes the authors have addressed both.
* Another potential limitation is that: the author assumes that the next subtask can be predicted given the video frames seen so far, which is generally true for short-horizon tasks. Consider a task with ambiguous order (e.g., put multiple objects into a box). The LLM might predict the robot to begin with object A first, making that the subtask, but the video might begin with object B.

**Quality:**

3

**Strengths And Weaknesses:**

# Quality

## Strengths

* The paper evaluates ROVER and baselines on 3 tasks, some testing an approach's ability to recognize progress and others testing the reasoning and understanding via QA. The newly introduced dataset exposes VLM's tendency to hallucinate success even when a video fail. And they also test on an existing benchmark for the progress prediction.
* Ablation tests the contribution of its two main component in the approach: (1) recursive algorithm, and (2) sliding window.
* The paper also tested the model's performance on various models.

## Weaknesses
* The paper is missing a baseline that performs local reasoning over a small set of frames. L217 mentions having LIV [1] as a baseline, but I cannot find any results in the main paper nor the appendix. If the author includes this baseline in rebuttal, I am willing to increase my score.
* Compared to other baselines (GVL and TemporalCat), ROVER is making more VLM calls as it recurses. What is the token cost for ROVER compared to other baselines?

[1] Ma, Y. J., Kumar, V., Zhang, A., Bastani, O., and Jayaraman, D. (2023). Liv: Language-image representations and rewards for robotic control. In International Conference on Machine Learning, pages 23301–23320. PMLR

# Clarity

## Strengths

*  The paper is well-written and well-organized with informative figures that clearly explain the approach.

# Significance & Originality

## Strengths

* The paper presents a new dataset with programmatically augmented subtoptimal demonstrations. Even though there might be a risk that these demonstrations are mismatched from suboptimal demonstrations in the real world, it is still interesting experiment results that the VLM tends to hallcinate success when there's suboptimal behavior.
* Figure 6 also presents interesting results that GVL which examines the video holistically tends to predict subtask occuring earlier compared to the ground-truth time.
* Results with TemporalCat supports GVL's claim that VLM has a tendency to predict increasing progress regardless of the input frames.

## Weaknesses
* The recursive algorithm does not seem to have a recovery mechanism. L114 says that the algorithm returns if the VLM determines the current subtask has been completed or if it reaches the end of the video. What happens when the spawned subtask is incorrect? In that case, a successful video would get marked as unsuccessful since that incorrect subtask fails to complete until the end of the video. What is authors' thought on mitigating this?
* Different approaches seem to require different compute and different number of LLM calls, making it difficult to compare them fairly.

---

> ### Author Rebuttal · Authors · 2025-07-29
>
> &nbsp;
> ### **Summary**
> We thank the reviewer for the comprehensive feedback. We are grateful for their recognition of our consistent performance improvements across diverse tasks, comprehensive experimental evaluation (for real-world and simulation) with ablations and different model backbones, and public dataset contribution.
>
> To address the reviewer’s questions and concerns, we make the following additions/changes:
>
> **(1)** We perform a comprehensive error analysis of ROVER (**Table R14**).
>
> **(2)** We add an evaluation of how the sliding window size affects ROVER’s performance (**Table R15**).
>
> **(3)** We add a baseline for local reasoning (**Tables R16 and R17**).
>
> **(4)** We add an analysis showing ROVER’s significant improvements in inference cost (**Tables R18 and R19**).
>
> **(5)** We provide deeper discussion of recovery mechanisms.
>
> &nbsp;
> ## **Questions**
> ### **Q1. Failure cases**
> To identify failure cases, we conducted an error analysis by hand reviewing 200 videos, calculating the rate of different errors (Table R14). Examples of each error type are shown in the response to Reviewer AqtN. The most common decomposition error type we observe with ROVER is specifying incorrect subtasks. For example, when decomposing the task “put the mug in the coffee machine”, the model might make an error by specifying the subtask of “open the cabinet door” when the mug is on the counter. The other decomposition error type to occur involved specifying redundant subtasks. For example, the model first decomposes the task “put the steak in the microwave” into the subtasks “pick up the steak from the counter” and “place the steak in the microwave”, but then incorrectly decomposes the  “place the steak in the microwave” subtask into “grasp the steak” and “move the steak to the microwave”.
>
> &nbsp;
>
> **Table R14. Error rates (%) observed among 100 real-world videos and 100 videos from simulation**
> | |**Real-world**|**Videos**| \| |**Simulation**|**Videos**|
> |:--|:--|:--|:--|:--|:--|
> |**Error type**|**GVL &darr;**|**ROVER &darr;**| \| |**GVL &darr;**|**ROVER &darr;**|
> |Incorrect subtasks|NA|9%| \| |NA|7%|
> |Redundant subtasks|NA|3%| \| |NA|4%|
> |Perception error|47%|7%| \| |32%|5%|
> |Reasoning error|14%|4%| \| |14%|4%|
> |Other|15%| 3%| \| |11%|3%|
> |Total|76%|26%| \| |57%|23%|
>
> &nbsp;
> &nbsp;
>
> ### **Q2. Trend in error rate**
> We appreciate the reviewer’s close reading of Figure 5. The observed fluctuations in error rate, where ROVER sometimes exhibits a slightly higher frame-level reasoning error even at higher levels of expertise (in pick-and-place and prepare-coffee tasks), can be attributed primarily to natural variance in the dataset and non-uniform difficulty across trajectories. The fluctuations in error rate are within the standard error margins and do not reflect a statistically significant trend in ROVER’s performance as trajectory quality improves. In addition, some of the higher-level trajectories still contain challenging edge cases. For instance, minimal deviations in timing or subtle failures in object manipulation may lead to increased reasoning difficulty even though the overall trajectory is successful. We will add deeper discussion of these trends in the revised paper.
>
> &nbsp;
> &nbsp;
>
> ### **Q3. Size of the sliding window**
> The sliding window includes 3 frames: the first frame of the current line of reasoning, the most recent previous frame, and the frame for the current timestep of the video. In Table R15 below, we show the performance of ROVER on the frame-level progress prediction task when changing the sliding window size to 6, 9, and 12 frames. We see that, as the number of frames included in the sliding window increases, the performance of ROVER drops. We will add these results to the revised paper.
>
> **Table R15.  Corr. with ground-truth values for frame-level progress prediction (with varying sliding context window size)**
> |Method|Corr. with Ground Truth &uarr;|
> |:--|:--|
> |ROVER (original)|**0.56**|
> |ROVER (window of 6)|0.52|
> |ROVER (window of 9)|0.50|
> |ROVER (window of 12)|0.44|
>
> &nbsp;
> &nbsp;
>
> ## **Quality**
> ### **W1. Adding baseline for local reasoning**
> We appreciate this feedback and agree that it is important to include a baseline for local reasoning. We include **LIV as a local-reasoning baseline for the frame-level progress prediction task**, as shown in the first results figure (Figure 3, page 6). Since LIV does not provide text outputs, we could not include it as a baseline for the frame-level natural language reasoning task or the video QA task. However, to include a reference for local reasoning for these tasks, **we add LocalConcat as a baseline** method in these evaluations, which is similar to TemporalConcat, but only includes the last three frames of the video in the model's context at each timestep. We provide the results of LocalConcat for frame-level reasoning (Table R16) and video QA (Table R17) below. We will add this to the main result section of the revised paper.
>
> **Table R16. Reasoning error rate for frame-level natural language reasoning (with extended baselines)**
> |Method|Error rate &darr;|
> |:--|:--|
> |ROVER|**45%**|
> |GVL|59%|
> |TemporalConcat|67%|
> |**LocalConcat (new)**|62%|
>
> &nbsp;
>
> **Table R17. Accuracy for video QA (with extended baselines)**
> |Method|Accuracy &uarr;|
> |:--|:--|
> |ROVER|**77%**|
> |GVL| 51%|
> |TemporalConcat|28%|
> |**LocalConcat (new)**|25%|
>
>
> &nbsp;
> &nbsp;
>
> ### **W2. Adding analysis of inference cost**
> The tables below show the inference cost per video for ROVER versus baselines. ROVER significantly decreases the average number of tokens per video, while slightly increasing the number of calls to the model (due to the recursive decomposition). For videos with 30 frames (the single-stage atomic tasks), ROVER shows about a **3x reduction** in the number of tokens per video (Table R18). For videos with 60 frames (the multi-stage composite tasks), ROVER shows more than a **5x reduction** in the number of tokens per video (Table R19). In the revised paper, we will include these results and a more thorough discussion of the observed improvements in inference cost with ROVER relative to baselines.
>
> **Table R18. Inference cost of each method for videos with 30 frames**
> |Method|Avg number of tokens per video|Avg number of VLM calls per video|
> |:--|:--|:--
> |ROVER|47,650|36|
> |GVL|140,307|30|
> |TemporalConcat|134,855|30|
>
> &nbsp;
>
> **Table R19. Inference cost of each method for videos with 60 frames**
> |Method|Avg number of tokens per video|Avg number of VLM calls per video|
> |:--|:--|:--
> |ROVER|95,404|68|
> |GVL|539,102|60|
> |TemporalConcat|527,572|60|
>
> &nbsp;
> &nbsp;
>
> ## **Significance & Originality**
> ### **W1. Recovery mechanism**
> As detailed in our response to Question 1 above, the most common decomposition errors we observe include errors associated with specifying incorrect subtasks or redundant subtasks. As you mention, if the model makes these errors, we do not enforce any explicit recovery mechanism. If the subtask is simply redundant and is already completed (e.g., stating the robot needs to grasp an object after it has already picked it up), the model will often recognize the redundant subtask (grasping the object) has already been completed and assign a subtask completion of 100%, causing it to move to the next subtask (placing the object) and proceed without an issue. If a subtask is incorrect, this generally causes the model to get stuck as you describe and fail in the overall video reasoning task. We discuss ideas for potential mitigation strategies below.
>
> **Mitigation strategies:** The most direct strategy for mitigating errors associated with specifying incorrect subtasks is to incorporate subtask validity constraints during attempted decompositions. Specifically, we can leverage task schemas or symbolic priors, derived from expert demonstrations or domain knowledge, to enforce that only logically and temporally valid subtasks are generated. Errors associated with specifying redundant subtasks could be mitigated with a subtask deduplication module, implemented via either clustering of subtask embeddings or attention-based novelty scoring. This module can flag subtasks that are semantically repetitive with recent ones and merge or skip them accordingly. We could also prevent the potential unstable recursive spiraling (which is often driven by repetitive subtask specifications) by preventing the model from further decomposing a subtask after a certain recursion depth is reached.
>
> &nbsp;
> &nbsp;
> ### **W2. Adding analysis of inference cost**
> We agree that it is important to consider inference cost when evaluating the performance of each method. Overall, ROVER **significantly reduces inference cost** relative to baselines. We include an analysis of the inference cost above (Tables R18 and R19). We see that, although ROVER slightly increase the number of calls to the model as it recurses, it dramatically decreases the number of tokens per video by shortening the number of frames the model must reason over at each timestep of the video (through the synergistic combination of recursive decomposition and a sliding context window). Further, since ROVER **scales linearly** with video length (while baselines **scale quadratically** with video length), the improvements in inference cost we observe with ROVER increase as video length increases.
>
> &nbsp;
> &nbsp;
> ## **Limitations**
> The issue of ambiguous subtask order is important and will be addressed in more detail in the limitations section. One solution is to explicitly specify task order (e.g., specifying the task as “place object A in box, then place object B in box” instead of “place objects A and B in box”). Alternatively, when that’s not feasible, the model can use order-agnostic phrasing, guided by appropriate prompting examples (e.g., given the task “place objects A and B in box”, the model can specify the first subtask as “place one of the objects in box” and the second subtask as “place the remaining object in box”).

---

> > ### Comment · Reviewer_JuQn · 2025-08-03
> >
> > I appreciate the author's detailed discussion. The author has addressed most of my concerns, but I have two more clarification questions:
> > - Why does ROVER perform drops in table R15, when the sliding window size increases? What are the failure cases induced by increased sliding window?
> > - Can you provide more explanation on how LocalConcat is implemented to do reasoning/QA tasks?

---

> ### Author Response · Authors · 2025-08-04
>
> &nbsp;
>
> Thank you for your follow-up questions! We include our answers below. Please let us know if any further results or explanations from us would be helpful.
>
> &nbsp;
> &nbsp;
>
> ### **Cause of performance drop when increasing the sliding window size.**
>
> To answer this question, we apply the same error analysis that we describe above (in our initial response to Question 1) to the modified versions of ROVER that have increasing sliding window size. Based on the results of this analysis shown in Table R20 below, the drop in performance with increasing window size appears to be driven by increases in perception errors, which also comprise the primary error type observed with GVL (as shown in Table R14 in our initial response). Perception errors occur when the model hallucinates basic facts about what is happening in the video (we include a full list of the error types with examples below). We see that, in the original version of ROVER, only 5% of videos include a perception error, but in the version of ROVER with a window size of 12, this rises to 19% of videos.
>
> This supports our findings that the VLM is prone to perception errors when attempting to reason over many frames of a video. ROVER (with a sliding window size of 3) mitigates these perception errors by shortening the number of frames the model must reason over at each timestep of the video. When the sliding window size is increased, perception errors increase. This is also consistent with our finding from the initial error analysis that 83% of perception errors from GVL occur when there are over 10 frames of the video included in the context.
>
> We see that these modified versions of ROVER still do not produce as many perception errors as GVL (which makes a perception error on over 30% of videos, as shown in Table R14). This is likely due to the fact that, even with a sliding window size of 12, ROVER still shortens the model’s context at each moment of reasoning by decomposing the video into subtask-specific lines of reasoning (which are generally less than 12 frames in length, even without implementing a sliding window).
>
> Overall, we find that ROVER introduces the possibility of decomposition errors, but dramatically reduces the **total error rate** by mitigating the model’s tendencies toward hallucination (perception errors). We will include Table R20, along with this discussion, in the revised paper.
>
> &nbsp;
>
> **Table R20.  Error rates (%) observed with increasing sliding window size**
> |Error type|ROVER (original)|ROVER (window of 6)|ROVER (window of 9)|ROVER (window of 12)|
> |:--|:--|:--|:--|:--|
> |Incorrect subtasks|7%| 6%| 5%| 7%|
> |Redundant subtasks|4%| 3%| 5%| 4%|
> |**Perception error**|5%| 10%| 12%| 19%|
> |Reasoning error|4%| 4%| 5%| 6%|
> |Other|3%| 2%| 4%| 4%|
> |Total|23%| 25%| 31%| 40%|
>
> &nbsp;
>
>
> **Error types with examples**
> - Incorrect subtasks
> 	- Example: When decomposing the task “put the mug in the coffee machine”, the model specifies the subtask of “open the cabinet door” when the mug is on the counter.
> - Redundant subtasks
> 	- Example: The model first decomposes the task “put the steak in the microwave” into “pick up the steak from the counter” and “place the steak in the microwave”, but then incorrectly decomposes the  “place the steak in the microwave” subtask into “grasp the steak” and “move the steak to the microwave”.
> - Perception error
> 	- Example: The model states that the gripper is holding an object when it is not.
> - Reasoning error
> 	- Example: Despite correctly identifying the gripper has dropped an object, the model states that task progress is increasing.
>
> &nbsp;
>
> &nbsp;
>
> &nbsp;
>
>
> ### **Deeper explanation of how LocalConcat is implemented for the reasoning and QA tasks.**
>
> LocalConcat is designed as a simple local reasoning baseline that mirrors the structure of TemporalConcat, but limits visual input to only a narrow temporal slice.
>
> For the frame-level reasoning task, LocalConcat includes the last 3 frames of the video in the VLM context at each timestep during reasoning. When generating the frame-level reasoning at timestep $t$, the model sees the frame at timesteps $t$, $t-1$, and $t-2$. The model also sees the frame-level reasoning it previously generated at timestep $t-1$ and $t-2$.
>
> Similar to all other methods, we evaluate LocalConcat for the video QA task by extracting the frame descriptions generated during the frame-level reasoning task and using a language model to generate an answer to each question given these frame descriptions concatenated together. Therefore, just like with the other methods, the QA performance provides a secondary measure that reflects the quality of the frame-level natural language reasoning of LocalConcat.
>
> Please let us know if you have any modifications you would like us to make to the implementation of LocalConcat or if there is another baseline method you would like us to add. We are happy to provide updated results with any modified methods that you would like to see.

---

> > ### Comment · Reviewer_JuQn · 2025-08-06
> >
> > I appreciate the author's follow-up responses, and they have addressed most of my concerns and questions. I will increase my score accordingly. I expect the detailed analysis (e.g., failure cases, inference cost) and additional baselines to be included in the final version of the paper.

---

> > > ### Author Response · Authors · 2025-08-06
> > >
> > > Thank you! And yes we will definitely include these additional analyses and baselines in the final version of the paper.

---

### Official Review · Reviewer_XMz4 · 2025-06-30

**Clarity:** 3
**Significance:** 4
**Originality:** 4
**Rating:** 5
**Confidence:** 4

**Summary:**

This paper proposes a recursive visual language model (VLM) reasoning framework designed for video sequences in embodied tasks. The framework enables step-by-step reasoning and enabling localized, temporally bounded reasoning. The authors evaluate their approach on a suite of generated benchmarks covering progress estimation, frame-level reasoning, and video question answering. Across all tasks, the proposed method demonstrates consistent improvements over baselines, highlighting its effectiveness in embodied reasoning.

**Questions:**

1. Can you provide clearer descriptions (and ideally a diagram) of the $\phi$ and $\psi$ functions? They are core to the method but currently hard to follow.
2. What do the axes in Figure 6 represent and how to analyze this figure? Also, would it be possible to present key numeric results in tables for easier comparison?
3. Can you include any simple embodied task experiments to better validate the framework in actual planning settings?

**Ethical Concerns:**

["NO or VERY MINOR ethics concerns only"]

**Final Justification:**

After the discussion with the authors during the rebuttal, my major concerns have been solved. I expect the further clarification (e.g., illustration figures and more detailed function clarifications) to be included in the final version of the paper.

**Limitations:**

yes

**Quality:**

4

**Strengths And Weaknesses:**

Strengths
1. The paper introduces a novel recursive reasoning approach that decomposes long-horizon reasoning tasks into subtasks, effectively reducing the burden on the model to retain extensive historical context. The motivation is straightforward and the solution sounds reasonable.
2. The authors devise a method to generate non-expert trajectories with ground-truth progress annotations from expert demonstrations. They also construct a dedicated benchmark for video reasoning in embodied scenarios, where the trajectories are enriched with fine-grained task progress signals. Extensive ablation and analysis are conducted to validate the proposed method on the benchmarks.
3. The proposed framework is evaluated on both automatically generated and real-world data. The results show that the method significantly reduces error rates and demonstrates fine-grained understanding of task completion states in embodied scenarios. Moreover, the approach exhibits a degree of generalization.

Weaknesses
1. The paper provides an unclear and somewhat abstract description of the core functions $\phi$ and $\psi$, particularly regarding their input-output and implementation details. A more explicit depiction—ideally in Algorithm 1 and potentially through visual illustrations—would greatly enhance clarity. Furthermore, the recursive nature of the framework raises concerns about error propagation and potential infinite loops in failure cases, which are not discussed in the paper.
2. Although the framework is designed for embodied tasks, the experimental evaluation is limited to video reasoning benchmarks. While these benchmarks test important capabilities, they fall short of establishing the framework’s effectiveness in a complete embodied planning and execution loop. This limits the conclusions that can be drawn about its applicability in real-world embodied settings.
3. The paper claims that the use of a sliding window results in linear time complexity with respect to video length. However, this point lacks thorough experimental validation or analysis. Additionally, many experiments rely on correlation with ground-truth as the evaluation metric. The non-absolute nature of this metric raises questions about its adequacy for evaluating task performance in a rigorous and comprehensive way.

---

> ### Author Rebuttal · Authors · 2025-07-29
>
> &nbsp;
> ### **Summary**
> We appreciate the reviewer’s detailed feedback. We are pleased they found that our work introduces an innovative technical approach with significant performance improvements, performs extensive evaluation on real-world data and simulation data, and offers a large new public dataset.
>
> To address the reviewer’s concerns, we make the following additions/changes:
>
> **(1)** We provide a clearer description of $\phi$ and $\psi$ functions.
>
> **(2)** We provider deeper discussion regarding the extension of this work for complete planning and execution loop.
>
> **(3)** We add an analysis showing ROVER’s linear time complexity and significant improvements in inference cost relative to baselines (**Tables R12 and R13**) and highlight expanded performance metics.
>
> &nbsp;
>
> ### **Q1. Descriptions of $\phi$ and $\psi$ functions**
>
> We fully agree that the methods section of our paper needs to provide clearer descriptions of the $\phi$ and $\psi$ functions and will update the revised paper accordingly. We will also include a more explicit depiction through visual illustrations in the method section. The **input** for the $\phi$ function includes all of the tokens generated by the parent line of reasoning. For example, in Figure 1, this includes all of the tokens leading up to the moment the model states “The robot needs to:...”. The **output** of the $\phi$ function is the new task description provided by the parent (based on what the model states the robot needs to do) along with the last frame of the parent line of reasoning. The **input** for the $\psi$ function includes all of the tokens generated by the child line of reasoning. For example, in Figure 1, this includes all of the tokens leading up to the moment the model states “Task is complete”. The **output** of the $\psi$ function is the final frame, and corresponding final text description, of the child process.
>
> &nbsp;
> &nbsp;
> ### **Q2. Figure 6 axes and providing tables**
> Figure 6 is a histogram where the **x axis** represents the difference in frames between when the VLM states that something occurred (e.g., the robot picks up an object) during a video and when it actually occurred during the video. For example, if the VLM states that the robot picked the object at frame 15 of the video, but the robot actually picks up the object at frame 20 of the video, the frame difference shown in Figure 6 would be -5 on the x-axis. If the VLM instead states that the robot pick up the object at frame 25, the frame difference shown in Figure 6 would be +5 on the x-axis. Therefore, negative frame differences in Figure 6 indicate the VLM is stating things occur before they actually occur during the video. The **y-axis** is simply the number of videos for which we observe each frame difference calculation. Overall, Figure 6 supports our findings that GVL has an hallucination problem where it frequently states things are occurring before they actually occur during the video. We will make sure to update x-axis and y-axis descriptions for Figure 6 to make this much clearer in the revised paper.
>
> &nbsp;
> &nbsp;
> ### **Q3 and W2. Embodied task experiments**
>
> We are excited to experiment with on-robot planning in ongoing/future work. The focus of this paper is primarily on benchmarking fine-grained task progress prediction and natural language reasoning given videos, but we fully agree that progress prediction has downstream applications in both planning and policy learning, and we are actively pursuing these directions as follow-up work.
>
> &nbsp;
> &nbsp;
> ### **W1. $\phi$ and $\psi$ functions and error propagation**
> We provide more thorough descriptions of the $\phi$ and $\psi$ functions (particularly their inputs and outputs) in response to Question 1 above. We will add clearer descriptions, along with a visual illustration, of these functions in the methods section of the revised paper.
>
> In our response to Reviewer AqtN, we conducted an error analysis by hand reviewing 200 videos, calculating the rate of different errors. The most common decomposition error type we observe with ROVER is specifying incorrect subtasks. For example, when decomposing the task “put the mug in the coffee machine”, the model might make an error by specifying the subtask of “open the cabinet door” when the mug is on the counter. The other decomposition error type to occur involved specifying redundant subtasks. For example, the model first decomposes the task “put the steak in the microwave” into the subtasks “pick up the steak from the counter” and “place the steak in the microwave”, but then incorrectly decomposes the  “place the steak in the microwave” subtask into “grasp the steak” and “move the steak to the microwave”.
>
> If the model makes these errors, we do not enforce any explicit recovery mechanism. If the subtask is simply redundant and is already completed (e.g., stating the robot needs to grasp an object after it has already picked it up), the model will often recognize the redundant subtask (grasping the object) has already been completed and assign a subtask completion of 100%, causing it to move to the next subtask (placing the object) and proceed without an issue. If a subtask is incorrect, this generally causes the model to get stuck as you describe and fail in the overall video reasoning task. We discuss ideas for potential mitigation strategies below.
>
> **Error mitigation strategies:** The most direct strategy for mitigating errors associated with specifying incorrect subtasks is to incorporate subtask validity constraints during attempted decompositions. Specifically, we can leverage task schemas or symbolic priors, derived from expert demonstrations or domain knowledge, to enforce that only logically and temporally valid subtasks are generated. Errors associated with specifying redundant subtasks could be mitigated with a subtask deduplication module, implemented via either clustering of subtask embeddings or attention-based novelty scoring. This module can flag subtasks that are semantically repetitive with recent ones and merge or skip them accordingly. We could also prevent the potential unstable recursive spiraling (which is often driven by repetitive subtask specifications) by preventing the model from further decomposing a subtask after a certain recursion depth is reached.
>
> &nbsp;
> &nbsp;
> ### **W3. Adding analysis of time complexity and expanded evaluation metrics**
>
> &nbsp;
>
> **Empirical validation of linear time complexity claim:** We appreciate the reviewer’s concern about the experimental support for our claim of linear time complexity with respect to video length. In the revised manuscript, we will provide a more detailed empirical analysis of inference cost as a function of video length. Specifically, we measure two key indicators of inference cost across tasks with different video lengths: (1) the total number of tokens processed per video and (2) the number of VLM calls. Tables R12 and R13 (below) show that ROVER consistently reduces token count compared to baseline methods, even though it requires a slightly higher number of model calls due to its recursive structure. Importantly, the increase in inference cost with respect to video length is approximately linear, consistent with our design goal. For instance, doubling the video length from 30 to 60 frames results in approximately a doubling of token usage for ROVER (from ~47k to ~95k tokens), supporting the linear scaling claim. Due to ROVER scaling linearly with video length (with baselines scaling quadratically), the improvements in inference cost we observe with ROVER increase as video length increases. For videos with 30 frames (the single-stage atomic tasks), ROVER shows about a 3x reduction in the number of tokens per video (Table R12). For videos with 60 frames (the multi-stage composite tasks), ROVER shows more than a 5x reduction in the number of tokens per video (Table R13). We will include these results and an expanded discussion in the revised manuscript to provide a clearer empirical basis for our time complexity claim.
>
> **Table R12. Inference cost of each method for videos with 30 frames**
> |Method|Avg number of tokens per video|Avg number of VLM calls per video|
> |:--|:--|:--
> |ROVER|47,650|36|
> |GVL|140,307|30|
> |TemporalConcat|134,855|30|
>
> &nbsp;
>
> **Table R13. Inference cost of each method for videos with 60 frames**
> |Method|Avg number of tokens per video|Avg number of VLM calls per video|
> |:--|:--|:--
> |ROVER|95,404|68|
> |GVL|539,102|60|
> |TemporalConcat|527,572|60|
>
> &nbsp;
>
> **Expanded performance metrics:** We agree that relying solely on Pearson correlation, a relative metric, for the progress prediction evaluation can be insufficient for comprehensive performance evaluation. To address this, in Appendix A (Figure 9), we include results across all task groups using an absolute error metric (L2 distance) for the progress prediction task. We will update the main results section of the revised paper to emphasize this. Further, we are happy to incorporate any additional performance metrics that you see fit.

---

> > ### Comment · Reviewer_XMz4 · 2025-08-04
> >
> > Thanks for the clarification of the reviewer's questions. Most of them have solved the concerns. There are still two main concerns.
> > 1) About the error propagation. Through the experiments from the author's rebuttal to other reviewers, the error mainly comes from the incorrect subtasks and the perception error. However, the authors did not provide any good solutions.
> > 2) Embodied task verification. The authors promise they will leave this for the future work without any simple test. While the reviewer thinks it is relatively important.
> > Except for these two limitations, the reviewer still thinks the paper is an interesting and solid work. So I maintain the original opinion.

---

> ### Author Response · Authors · 2025-08-04
>
> &nbsp;
>
> We thank the reviewer for their thoughtful follow-up and for acknowledging the overall strength and interest of our work.
>
> &nbsp;
>
> ### **(1) On solutions for error propagation**
> While we do not yet implement dedicated mitigation modules in the current version of ROVER (to maintain minimal architectural complexity and isolate the core contribution), we have developed concrete mitigation strategies that can be integrated in future iterations, which we outline further below. Importantly, despite not yet implementing these strategies, ROVER already significantly outperforms existing baselines in terms of both accuracy and efficiency. This reflects the method’s robustness in practice, even under imperfect decomposition (although adding mitigation strategies is a promising direction for future refinement).
>
>
> Error propagation mitigation solutions
> - **Validity constraints**: Enforcing task schemas or commonsense constraints to prevent invalid or incorrect subtask sequences.
> - **Subtask deduplication**: Using semantic embedding similarity or attention-based novelty scoring to filter redundant subtasks.
> - **Recursion depth limits**: Setting a maximum recursion depth to prevent unstable or infinite decompositions.
> - **Self-checkpointing**: Integrating subtask verification loops using auxiliary progress predictors or visual affordance models.
>
>
> &nbsp;
>
> &nbsp;
>
> ### **(2) On embodied task verification**
> We fully agree that real-world embodied execution is an important long-term application of video-based reasoning.
> Specifically, we are integrating ROVER with visual policy learning pipelines where predicted progress and subtasks guide modular policy selection and execution. These results will be reported in follow-up work, where we can more comprehensively evaluate robustness to perception noise, embodiment constraints, and control latency.
>
> &nbsp;
>
> &nbsp;
>
> Once again, we sincerely thank the reviewer for their constructive feedback. We are encouraged that the core contributions are viewed as technically solid and impactful, and we look forward to building on this foundation with future work in physical settings and with improved robustness mechanisms.

---

### Official Review · Reviewer_Y85s · 2025-07-01

**Clarity:** 3
**Significance:** 2
**Originality:** 3
**Rating:** 3
**Confidence:** 3

**Summary:**

This paper proposes ROVER, a new vision-language model (VLM) video reasoning framework designed to enhance model understanding and reasoning capabilities for long-sequence videos in embodied intelligence tasks. The core innovation lies in recursively decomposing lengthy task videos into shorter, more semantically meaningful sub-task segments, enabling the model to perform more focused and accurate reasoning for each sub-task. The framework introduces a sub-task-level sliding window mechanism that considers only the most relevant frames for the current sub-task at each step, significantly reducing computational complexity and effectively mitigating model "hallucination" problems in abnormal or sub-optimal states. The authors implement ROVER using in-context learning and conduct systematic evaluation on large-scale datasets containing both expert and non-expert (perturbed) robot manipulation videos, establishing three benchmark tasks: frame-by-frame task progress estimation, frame-by-frame natural language reasoning, and video question answering.

**Questions:**

1. **Task decomposition failure analysis**: You acknowledge that reasoning may become fragmented when decomposition fails but lack systematic analysis. Could you provide specific cases and failure modes of task decomposition failures, and discuss potential mitigation strategies?

2. **Simulation-to-real transfer**: Have you conducted any validation for transferring from simulation to real-world scenarios?

3. **Video length performance analysis**: The paper doesn't seem to address performance differences between short and long videos. In task planning, different task types exist (simple, complex, common, long-horizon). Could you supplement experiments comparing ROVER's relative advantages on short vs. long videos?

**Ethical Concerns:**

["NO or VERY MINOR ethics concerns only"]

**Limitations:**

The authors acknowledge that data generated based on RoboCasa simulation environment may have biases affecting real-world generalization. While mentioning linear scaling, the actual computational overhead of recursive calls and multiple VLM inferences may still be high.

**Quality:**

3

**Strengths And Weaknesses:**

**Strengths:**
- **Comprehensive experimental evaluation**: Experiments conducted on both self-built RoboCasa dataset and public OpenX Embodiment dataset, covering three major tasks (task progress estimation, frame-by-frame reasoning, and video QA) with convincing results
- **Consistent performance improvements**: ROVER significantly outperforms strong baseline methods across all benchmark tasks, particularly excelling in reducing model hallucination and improving reasoning accuracy and efficiency
- **Valuable dataset contribution**: Provides high-quality, diverse expert/non-expert robot video datasets by generating varied non-expert trajectories through expert demonstration perturbation
- **Algorithmic efficiency**: Linear scaling with video length (compared to quadratic scaling of baselines) represents an important algorithmic improvement
- **Novel technical approach**: Combines recursive decomposition with sub-task sliding windows, solving the trade-off between losing global information and excessive computational cost

**Weaknesses:**
- **Limited real-world validation**: Experiments are primarily based on simulation and annotated data; while appendix mentions real-world dataset results, main evaluation remains limited to RoboCasa simulation environment
- **Task decomposition dependency**: Success relies heavily on correct task decomposition; lacks discussion of failure cases when decomposition introduces unnecessary or invalid sub-tasks
- **Limited baseline comparisons**: While comparing with GVL and TemporalConcat, lacks comparison with more recent video reasoning methods, particularly those designed for long videos
- **Insufficient failure analysis**: Acknowledges that reasoning may become fragmented when decomposition fails but lacks systematic analysis of failure modes

---

> ### Author Rebuttal · Authors · 2025-07-29
>
> &nbsp;
> ### **Summary**
> We thank the reviewer for their thoughtful feedback. We are grateful for their recognition of our comprehensive experimental evaluation, novel technical advancement, significant performance improvements across tasks, and public dataset contribution.
>
> To address the reviewer’s questions and concerns, our primary changes include the following:
>
> **(1)** We perform a comprehensive error analysis for both real-world videos and videos from simulation  (**Table R6**).
>
> **(2)** We add deeper context and discussion for our extensive real-world validation (**Table R7**), including clear demos of ROVER’s improved real-world reasoning.
>
> **(3)** We add 3 more baselines based on recent methods designed for video understanding for our real-world and sim evaluation (**Tables R8, R9, R10**).
>
> **(4)** We add an analysis of inference cost (**Table R11**).
>
> &nbsp;
>
> ### **Q1. Failure analysis and mitigation strategies**
> We hand reviewed 100 real-world videos and 100 videos from simulation to identify the most common types of errors (Table R6) made by ROVER and GVL. Below we outline the full list of error types and provide specific example cases for each.
>
> The most common error type for ROVER is specifying incorrect subtasks, where 9% of real-world videos and 7% of simulation videos include at least one occurrence of this error. The most common error type for GVL is perception errors, where 47% of real-world videos and 32% of simulation videos include at least one occurrence of this error. Further, we observe that 83% of perception errors from GVL occur when there are over 10 frames of the video included in the context. This supports our findings that the VLM is prone to perception errors, such as hallucination, when attempting to reason over many frames of a video. ROVER mitigates these perception errors by shortening the number of frames the model must reason over at each timestep of the video (through the synergistic combination of recursive decomposition and a sliding context window). **Overall, we find that ROVER introduces the possibility of decomposition errors, but dramatically reduces the total error rate by mitigating the model’s tendencies toward hallucination.**
>
> **Error types and examples**
> - Incorrect subtasks
> 	- Example case: When decomposing the task “put the mug in the coffee machine”, the model specifies the subtask of “open the cabinet door” when the mug is on the counter.
> - Redundant subtasks
> 	- Example case: The model first decomposes the task “put the steak in the microwave” into “pick up the steak from the counter” and “place the steak in the microwave”, but then incorrectly decomposes the  “place the steak in the microwave” subtask into “grasp the steak” and “move the steak to the microwave”.
> - Perception error
> 	- Example case: The model states that the gripper is holding an object when it is not.
> - Reasoning error
> 	- Example case: Despite correctly identifying the gripper has dropped an object, the model states that task progress is increasing.
>
> &nbsp;
>
> **Table R6. Error rates (%) observed among 100 real-world videos and 100 simulation videos**
> | |**Real-world**|**Videos**| \| |**Simulation**|**Videos**|
> |:--|:--|:--|:--|:--|:--|
> |**Error type**|**GVL &darr;**|**ROVER &darr;**| \| |**GVL &darr;**|**ROVER &darr;**|
> |Incorrect subtasks|NA|9%| \| |NA|7%|
> |Redundant subtasks|NA|3%| \| |NA|4%|
> |Perception error|47%|7%| \| |32%|5%|
> |Reasoning error|14%|4%| \| |14%|4%|
> |Other|15%| 3%| \| |11%|3%|
> |Total|76%|26%| \| |57%|23%|
>
> &nbsp;
>
> **Mitigation strategies:** To avoid errors in task decomposition, correct subtasks can be ensured using constraints based on expert-derived schemas, while redundant subtasks can be reduced through deduplication modules using embedding clustering or novelty scoring. We include deeper discussion of mitigation strategies in response to Reviewer JuQn under “Recovery mechanism”.
>
> &nbsp;
> &nbsp;
>
> ### **Q2. Simulation-to-real transfer**
> To test the effectiveness of ROVER in real-world scenarios, we test ROVER on the largest collection of real-world robotics datasets publicly available, the Open X-Embodiment (OXE) datasets. As you mention, we initially include this analysis in the appendix, but we will add the extended analysis to the main results section of the revised paper to ensure it is properly highlighted. We also include clear demonstrations of ROVER’s improved reasoning on real-world videos in Appendix K and include the real-world videos in the error analysis above.
>
> Overall, compared to GVL, ROVER shows significantly higher correlation (Tables 4 and 5 in Appendix K and Table R7 below) with ground-truth values among known high-quality datasets collected from human teleoperators with fixed camera placements (e.g., Bridge, RT-1, and Dobb-E). The OXE videos include 20 robot embodiments, over 300 different task specifications, and highly diverse (often non-ideal) camera viewpoints and environmental conditions. **To our knowledge, this is among the most comprehensive real-world VLM evaluations conducted to date across such a heterogeneous set of robot platforms and data sources.** The significant improvement in performance achieved by ROVER on these datasets demonstrates its ability to generalize (zero-shot) to real-world conditions (including variations in lighting, clutter, occlusion, and partial observability).
>
>
>
> **Table R7. Examples from real-world results**
> | Real-world dataset | GVL Corr. with Ground Truth &uarr;| ROVER Corr. with Ground Truth &uarr;|
> |:--|:--|:--|
> |Bridge |0.457|**0.717**|
> |RT-1| 0.762|**0.817**|
> |Dobb-E|0.487|**0.754**|
> |Berkeley Cable Routing |0.488|**0.752**|
>
> &nbsp;
> &nbsp;
>
> ### **Q3. Video length performance analysis**
> We fully agree it is interesting and relevant to evaluate performance differences between short and long videos. We included this analysis in Appendix H, but will add discussion of these results in the main results section. **Overall, we see that ROVER is less sensitive to video length and video frame rate relative to GVL and TemporalConcat.** We first observe this in the results stratified across task group (Figures 3 and 5 from the main results section), where we see the improved performance of ROVER for the composite tasks (bottom row of each figure), which include much longer videos than the atomic tasks (top row of each figure). We also observe ROVER’s robustness to frame rate when varying the total number of frames used to represent each video between 30 and 240 for the pick-and-place tasks for the progress prediction task (Figure 18 in Appendix H) and frame-level natural language reasoning (Figure 19 in Appendix H). **These figures show that the performance of GVL and TemporalConcat significantly degrades when given the same pick-and-place videos, but with effective frame rate increased.**
>
> &nbsp;
> &nbsp;
>
> ### **W1. Limited real-world validation**
> We agree that we should better emphasize our extensive evaluation on real-world datasets in the main results section, which we detail in our response to Question 2 above. We will revise the main results section to highlight these real-world results more prominently, including performance breakdowns by dataset quality and embodiment diversity.
>
> &nbsp;
> &nbsp;
>
> ### **W2. Task decomposition dependency and failure analysis**
> We include the decomposition failure analysis as part of the comprehensive error analysis in our response to Question 2 above.
>
> &nbsp;
> &nbsp;
>
> ### **W3. Limited baseline comparisons**
> We fully agree that it is important to consider recent methods and models tailored for video understanding. **We have added two recent video-specific baselines: VideoLLaMA-3 and Video-LLaVA.** These models are designed specifically for video-language tasks and incorporate temporal modeling and memory mechanisms that target long-sequence video comprehension.  **We also add the baseline VideoGemini** that concatenates all frames of the video to provide as input to the model, more closely resembling the manner in which Gemini was trained for video reasoning tasks. We provide the updated results below for both real-world videos and videos from simulation. We will update the main results section of the paper to include these baselines. If you have any other methods or models in mind that you would like us to include as baselines, we would be happy to add them.
>
>
> **Table R8.  Corr. with ground-truth values for frame-level progress prediction (with extended baselines)**
> |Method|Real-world Videos Corr. &uarr;|Simulation Videos Corr. &uarr;|
> |:--|:--|:--|
> |ROVER|**0.62**|**0.56**|
> |GVL|0.31|0.35|
> |TemporalConcat |0.43|0.17|
> |LIV|0.12|0.09|
> |**VideoGemini (new)**|0.46|0.29|
> |**VideoLlama3 (new)**|0.34|0.21|
> |**VideoLLaVA (new)**|0.19|0.14|
>
> &nbsp;
>
> **Table R9. Reasoning error rate for frame-level natural language reasoning (with extended baselines)**
> |Method|Error rate &darr;|
> |:--|:--|
> |ROVER|**45%**|
> |GVL|59%|
> |TemporalConcat|67%|
> |**VideoGemini (new)**|62%|
> |**VideoLlama3 (new)**|71%|
> |**VideoLLaVA (new)**|76%|
>
> &nbsp;
>
> **Table R10. Accuracy for video QA (with extended baselines)**
> |Method|Accuracy &uarr;|
> |:--|:--|
> |ROVER|**77%**|
> |GVL| 51%|
> |TemporalConcat|28%|
> |**VideoGemini (new)**|68%|
> |**VideoLlama3 (new)**|62%|
> |**VideoLLaVA (new)**|58%|
>
> &nbsp;
> &nbsp;
> ### **W4. Insufficient failure analysis**
> We include a full error analysis in our response to Question 2 above.
>
> &nbsp;
> &nbsp;
>
> ### **Limitation: Computational overhead**
> The table below shows inference cost per video for ROVER versus baselines. We see that ROVER significantly decreases the average number of tokens per video, despite slightly increasing the number of calls to the model (due to the recursive decomposition). We will include a more thorough discussion of the inference cost in the revised paper.
>
> **Table R11. Inference cost of each method for videos with 30 frames**
> |Method|Avg number of tokens per video|Avg number of VLM calls per video|
> |:--|:--|:--
> |ROVER|47,650|36|
> |GVL|140,307|30|
> |TemporalConcat|134,855|30|

---

### Official Review · Reviewer_AqtN · 2025-07-03

**Clarity:** 3
**Significance:** 3
**Originality:** 3
**Rating:** 3
**Confidence:** 3

**Summary:**

This paper introduces ROVER, a recursive vision-language framework designed for embodied robotic tasks. It enhances reasoning by decomposing long-horizon videos into manageable subtasks. At its core, ROVER recursively segments task trajectories to maintain global context, while a sliding window within each subtask focuses the model on temporally relevant frames. This dual approach allows it to effectively balance high-level task structure with fine-grained local details. Evaluated on a new, diverse RoboCasa dataset and other large-scale benchmarks, ROVER significantly outperforms leading baselines (GVL, TemporalConcat, LIV) in task progress estimation, frame-level reasoning, and video QA. Notably, it mitigates model hallucinations and achieves linear time complexity, scaling efficiently with video length.

**Questions:**

See above.

**Ethical Concerns:**

["NO or VERY MINOR ethics concerns only"]

**Final Justification:**

After the discussion with the authors during the rebuttal, some of my concerns have been solved. However, I still believe that the task progress estimator was only trained on table-top video datasets. Like Reviewer XMz4, I have the same concern about ROVER's performance in real-world downstream task, which could be explored in future work.

**Limitations:**

Yes.

Limitations are adequately acknowledged (Section 7 and Broader Impact), with constructive discussion on decomposition failure, reliance on in-context prompting, and the risk of increased autonomy in AI systems.

**Paper Formatting Concerns:**

No.

**Quality:**

3

**Strengths And Weaknesses:**

Strengths:

1. Clear and Compelling Motivation: The work is well-motivated, addressing the fundamental trade-off in VLM-based video reasoning between maintaining global context and achieving computational efficiency.
2. Innovative Methodology: The core contribution is ROVER’s innovative methodology, which synergistically combines recursive task decomposition with a sliding context window. This elegant approach simultaneously enables fine-grained, focused reasoning while maintaining computational efficiency.
3. Rigorous and Extensive Experiments: The claims are supported by a rigorous empirical evaluation using a novel, large-scale RoboCasa dataset alongside established benchmarks. This provides robust validation across diverse and realistic manipulation tasks, strengthening the generalizability of the findings.
4. Valuable Community Resource: The public release of the dataset, code, and demonstrations constitutes a valuable contribution to the research community.

Weaknesses:

1. Lack of Decomposition Failure Analysis: The framework's performance heavily relies on successful task decomposition, yet the paper lacks a quantitative analysis of potential failure modes. The frequency and impact of incorrect subtask splits are not explored, making it difficult to assess the method's robustness.
2. Uncertain Generalization Scope: The primary contributions and in-depth analysis are centered on simulated robotic manipulation. The framework's applicability to other challenging video domains, such as open-world videos or real-world scenarios, remains undemonstrated.
3. Limited Set of Modern Baselines: The comparison could be strengthened by including more recent architectures designed for long-sequence video understanding, such as advanced transformer variants or memory-augmented VLMs, which would provide a more competitive performance benchmark.
4. Potential Dependence on Task Structure: The assumption that tasks can be decomposed into object-centric subtasks with clear sequential transitions (as in RoboCasa) may not hold for more entangled or ambiguous real-world tasks. The implications or limits of this assumption warrant deeper discussion, beyond referencing previous work.

---

> ### Author Rebuttal · Authors · 2025-07-29
>
> &nbsp;
> ### **Summary**
> We appreciate the reviewer’s detailed feedback. We are pleased they found that our work has a clear and compelling motivation, introduces an innovative technical approach, offers rigorous experimental validation, and contributes a significant public dataset.
>
> To address the reviewer’s concerns, we make the following additions/changes:
>
> **(1)** We perform a comprehensive error analysis for both real-world videos and videos from simulation  (**Table R1**).
>
> **(2)** We add deeper analysis and discussion for extensive real-world experiments (**Table R2**), including clear demonstrations of ROVER’s improved reasoning on real-world examples.
>
> **(3)** We add 3 more modern baselines that are designed for video understanding for our real-world and sim evaluation (**Tables R3, R4, R5**).
>
> **(4)** We add more extensive inspection to provide real-world examples of effective reasoning with tasks without standard object-centric subtasks.
>
> &nbsp;
> ### **W1. Lack of Decomposition Failure Analysis**
> We appreciate this feedback and agree it is critical to include an analysis of the types of errors made by ROVER and GVL. To address this, we hand-reviewed 100 real-world videos and 100 videos from simulation to identify the most common types of errors for each method (Table R1). Below we outline the full list of error types and provide examples for each. We will include this analysis in the main results section of the revised paper.
>
> **Findings:** The most common error type for ROVER is specifying incorrect subtasks, where 9% of real-world videos and 7% of simulation videos include at least one occurrence of this error. The most common error type for GVL is perception errors, where 47% of real-world videos and 32% of simulation videos include at least one occurrence of this error. Further, we observe that 83% of perception errors from GVL occur when there are over 10 frames of the video included in the context. This supports our findings that the VLM is prone to perception errors, such as hallucination, when attempting to reason over many frames of a video. ROVER mitigates these perception errors by shortening the number of frames the model must reason over at each timestep of the video (through the synergistic combination of recursive decomposition and a sliding context window). **Overall, we find that ROVER introduces the possibility of decomposition errors, but dramatically reduces the total error rate by mitigating the model’s tendencies toward hallucination.**
>
> **Error types with examples**
> - Incorrect subtasks
> 	- Example: When decomposing the task “put the mug in the coffee machine”, the model specifies the subtask of “open the cabinet door” when the mug is on the counter.
> - Redundant subtasks
> 	- Example: The model first decomposes the task “put the steak in the microwave” into “pick up the steak from the counter” and “place the steak in the microwave”, but then incorrectly decomposes the  “place the steak in the microwave” subtask into “grasp the steak” and “move the steak to the microwave”.
> - Perception error
> 	- Example: The model states that the gripper is holding an object when it is not.
> - Reasoning error
> 	- Example: Despite correctly identifying the gripper has dropped an object, the model states that task progress is increasing.
>
> &nbsp;
>
> **Table R1. Error rates (%) observed among 100 real-world videos and 100 videos from simulation**
> | |**Real-world**|**Videos**| \| |**Simulation**|**Videos**|
> |:--|:--|:--|:--|:--|:--|
> |**Error type**|**GVL &darr;**|**ROVER &darr;**| \| |**GVL &darr;**|**ROVER &darr;**|
> |Incorrect subtasks|NA|9%| \| |NA|7%|
> |Redundant subtasks|NA|3%| \| |NA|4%|
> |Perception error|47%|7%| \| |32%|5%|
> |Reasoning error|14%|4%| \| |14%|4%|
> |Other|15%| 3%| \| |11%|3%|
> |Total|76%|26%| \| |57%|23%|
>
> &nbsp;
> &nbsp;
>
> ### **W2. Uncertain Generalization Scope**
> We agree that it is important to evaluate our method on real-world videos. We test ROVER on the largest collection of real-world robotics datasets publicly available, the Open X-Embodiment (OXE) datasets. The OXE dataset includes an aggregation of trajectory data from 50 standalone robot datasets that consists of diverse tasks, robots, and camera viewpoints. For each of the 50 datasets, we randomly sample 20 trajectories and evaluate ROVER zero-shot on each of the sampled trajectories. The OXE videos include 20 robot embodiments, over 300 different task specifications, and highly diverse (often non-ideal) camera viewpoints. The significant improvement in performance achieved by ROVER (results described below and real-world demonstrations are included in Appendix K) on these datasets demonstrates its ability to generalize (zero-shot) to real-world conditions, including variations in lighting, clutter, occlusion, and partial observability. **To our knowledge, this is among the most comprehensive real-world VLM evaluations conducted to date across such a heterogeneous set of robot platforms and data sources.** We originally included these results in the appendix, but we will update the main results section to highlight this significant analysis, including performance breakdowns by dataset quality and embodiment diversity. Further, we include the real-world videos in the error analysis above, which will also be added to the main results section.
>
> **Findings:** We include tables (Tables 4 and 5) in Appendix K that show the average correlation each method achieves with the ground-truth value estimates across the OXE datasets when using Gemini-1.5-Pro and GPT-4o as the backbone VLM. We also include clear demonstrations of ROVER’s improved reasoning on real-world videos in Appendix K. For known high-quality datasets collected from human teleoperators with fixed camera placements, such as Bridge, RT-1, and Dobb-E, ROVER achieves a significantly higher correlation with frame number than GVL (highlighted in yellow in the Appendix K tables and shown below in Table R2).
>
> **Table R2. Examples from real-world results**
> | Real-world dataset | GVL Corr. with Ground Truth &uarr;| ROVER Corr. with Ground Truth &uarr;|
> |:--|:--|:--|
> |Bridge |0.457|**0.717**|
> |RT-1| 0.762|**0.817**|
> |Dobb-E|0.487|**0.754**|
> |Berkeley Cable Routing |0.488|**0.752**|
>
> &nbsp;
> &nbsp;
>
> ### **W3. Limited Set of Modern Baselines**
> We fully agree that a comprehensive evaluation should consider recent models tailored for video understanding. **We have added two recent video-specific baselines: VideoLLaMA-3 and Video-LLaVA.** These models are designed specifically for video-language tasks and incorporate temporal modeling and memory mechanisms that target long-sequence video comprehension. **We also add a baseline, VideoGemini,** that concatenates all frames of the video to provide as input to the model, more closely resembling the manner in which Gemini was trained for video reasoning tasks. If you have any other models or methods in mind that you would like us to include as baselines, we would be happy to add them.
>
> We provide the updated results below for both real-world videos and videos from simulation. Our results indicate that SOTA video reasoning models underperform ROVER in task progress prediction, frame-level reasoning, and video question answering. We hypothesize that recursive task decomposition in ROVER is a highly beneficial prior for reasoning about embodied tasks. We further find that video reasoning models are relatively weaker at frame-level progress prediction and reasoning compared to video question answering, which indicates that there is room for improvement in reasoning about fine-grained robot-object interactions. We will include these results in the main results section of the revised paper.
>
> &nbsp;
> &nbsp;
>
> **Table R3.  Corr. with ground-truth values for frame-level progress prediction (with extended baselines)**
> |Method|Real-world Videos Corr. &uarr;|Simulation Videos Corr. &uarr;|
> |:--|:--|:--|
> |ROVER|**0.62**|**0.56**|
> |GVL|0.31|0.35|
> |TemporalConcat |0.43|0.17|
> |LIV|0.12|0.09|
> |**VideoGemini (new)**|0.46|0.29|
> |**VideoLlama3 (new)**|0.34|0.21|
> |**VideoLLaVA (new)**|0.19|0.14|
>
> &nbsp;
>
> **Table R4. Reasoning error rate for frame-level natural language reasoning (with extended baselines)**
> |Method|Error rate &darr;|
> |:--|:--|
> |ROVER|**45%**|
> |GVL|59%|
> |TemporalConcat|67%|
> |**VideoGemini (new)**|62%|
> |**VideoLlama3 (new)**|71%|
> |**VideoLLaVA (new)**|76%|
>
> &nbsp;
>
> **Table R5. Accuracy for video QA (with extended baselines)**
> |Method|Accuracy &uarr;|
> |:--|:--|
> |ROVER|**77%**|
> |GVL| 51%|
> |TemporalConcat|28%|
> |**VideoGemini (new)**|68%|
> |**VideoLlama3 (new)**|62%|
> |**VideoLLaVA (new)**|58%|
>
> &nbsp;
> &nbsp;
>
> ### **W4. Potential Dependence on Task Structure**
>
> We agree that object-centric subtask decomposition warrants further discussion. Firstly, we would like to distinguish between data generation assumptions and methodological assumptions. In order to generate our simulation benchmark data, we leveraged the data replication technique introduced in MimicGen, which indeed assumes that we can decompose a task into object-centric subtasks. This is used to generate ground-truth task progress values.
>
> On the other hand, our proposed method, ROVER, does not place strict assumptions on the nature of each subtask. For example, ROVER can predict the task progress for cable routing (Berkeley Cable Routing, Table R3) with high correlation to the ground-truth, even though the task of clipping a cable into a slot cannot be easily described with rigid-body pick-and-place physics. Upon deeper inspection of reasoning traces for this task, we see that ROVER effectively reasons about the gripper “angling the cable toward the slots” observed during the video. We will add these real-world example demonstrations in the revised paper and incorporate this deeper discussion. In general, ROVER assumes that the top-level task has semantically meaningful subtasks, but the nature of these subtasks is limited only by the capabilities of the base vision-language model.

---

> > ### Author Response · Authors · 2025-08-06
> >
> > We wanted to follow up to check if you’ve had a chance to review our response. We’ve added several new analyses and experiments (outlined above) that we believe directly address your concerns, particularly around decomposition failure analysis, demonstrating real-world generalization, extended baseline comparisons, and validating on tasks without clear object-centric subtasks. Please let us know if you have any questions or if there is anything further we can do to resolve your concerns.

---

> ### Author Response · Authors · 2025-08-07
>
> > *For a deployed robotic system, the main concern is whether it can execute a task, rather than how accurately it can provide a post-hoc commentary on its own performance.*
>
> We agree that task execution is the primary goal of any deployed robotic system. However, achieving robust, general-purpose task execution requires more than just an action policy. It also demands introspection, monitoring, and the ability to reason about ongoing performance.
>
> Task progress estimation is not merely “post-hoc commentary.” It is a critical capability that enables:
> - Failure detection in dynamic or unpredictable environments
> - Assistance-seeking behaviors when a robot stalls or deviates from the intended plan
> - Re-planning in long-horizon tasks with multiple stages or contingencies
> - Self-improvement through feedback in reinforcement learning
>
> These capabilities are essential for real-world deployments, where sensory noise, unstructured environments, and unexpected events are the norm. A purely reactive policy, no matter how well-trained, cannot robustly handle the variability and complexity of the real world without some mechanism for evaluating and adjusting its own behavior.
>
> Our work directly supports these goals by providing a scalable method for fine-grained, frame-level progress estimation from raw video. This allows robots to monitor their own performance using onboard sensors, without relying on task-specific instrumentation, which is crucial for autonomy at scale.
>
> &nbsp;
>
> > *Therefore, the research problem addressed in this work appears somewhat disconnected from the core challenges of the field.*
>
> This assertion is not supported by the facts. As we’ve outlined, video-based reasoning and task understanding are key challenges in embodied AI, particularly as VLMs are increasingly used to interface with robotic systems. You even highlight the **“clear and compelling motivation”** of our work as a key strength of the paper **in your original review**.  Our paper directly targets this problem and, as you mention in your review, contributes meaningful advances through a scalable, focused reasoning framework.
>
> &nbsp;
>
> > *Potential Limitations of the Method in Complex Navigation Tasks*
>
> This criticism is a non sequitur. Our method does not attempt to solve navigation tasks, nor do we claim to. The focus of our work is video reasoning in manipulation-centric scenarios. Attempting to evaluate our approach in long-range navigation settings is not only outside the scope of this work but also misaligned with the objectives and framing of our submission.
>
> &nbsp;
>
> > *Although the paper claims its method is robust to camera view changes, this conclusion was likely drawn from its simplified, "navigation-light" task settings and it remains a significant question whether this robustness would generalize to more realistic, navigation-intensive household tasks.*
>
> We don’t make any claims about navigation. We don’t do navigation.
>
> &nbsp;
>
> &nbsp;
>
> **References**
>
> [1] Grasp Learning: Models, Methods, and Performance. Robert Platt. Annual Review of Control, Robotics, and Autonomous Systems. 2022.
>
> [2] OpenVLA: An Open-Source Vision-Language-Action Model. Kim et al. CoRL. 2025
>
> [3] Octo: An Open-Source Generalist Robot Policy. Ghosh et al. RSS. 2024
>
> [4] Do As I Can, Not As I Say: Grounding Language in Robotic Affordances. Ahn et al. CoRL. 2023.
>
> [5] PaLM-E: An Embodied Multimodal Language Model. Driess et al. ICML. 2023.
>
> [6] Policy invariance under reward transformations. Ng et al. ICML. 1999.
>
> [7] The Ingredients of Real-World Robotic Reinforcement Learning. Zhu et al. ICLR. 2020.
>
> [8] Deep Reinforcement Learning for Robotics: A Survey of Real-World Successes. Tang et al. Annual Review of Control, Robotics, and Autonomous Systems. 2024.

---

> > ### Comment · Reviewer_AqtN · 2025-08-08
> >
> > I understand your work focuses on table-top tasks without navigation, and recognize the significance of reward shaping. Although the proposed method is novel and effective for table-top scenarios, the task progress estimator was only trained on table-top video datasets. Like Reviewer XMz4, I have the same concern about ROVER's performance in real-world downstream task, which could be explored in future work.

---

> > > ### Author Response · Authors · 2025-08-08
> > >
> > > Thank you for your follow up. We would just like to clarify that ROVER is based on an in-context learning approach and **does not involve any training of the model on new data**. Also, we validate ROVER by testing on over 300 tasks from the real-world OXE datasets and 27 tasks in simulation, which **extend far beyond table-top tasks**. For example, the **RT-1 dataset** [1] in OXE (referenced in **Table R2** of our initial response highlighting ROVER’s strong real-world performance) includes a wide variety of real-world mobile manipulation tasks, such as unloading items from a dishwasher, moving objects into different containers, and positioning tools or utensils, many of which involve mobile multi-step interactions with the environment. In addition, the **Dobb-E dataset** [2] (also in **Table R2**) consists of real-home long-horizon tasks involving complex mobile manipulation, such as fetching objects, loading laundry machines, or preparing food items. The RoboCasa dataset [3] includes opening/closing cabinet doors and drawers, turning sink spouts and handles, multi-step pick-and-place tasks, and twisting knobs (along with 5 long-horizon composite tasks).
> > >
> > > We appreciate that you and Reviewer XMz4 share a similar concern regarding downstream applications of this work, and note that Reviewer XMz4 ultimately gave our paper a positive rating. Given ROVER’s strong performance on both real-world and sim benchmarks, we agree it has compelling downstream applications, which we are excited to be exploring as part of ongoing/future work. We have directly addressed your original concerns and kindly ask that you consider increasing your score.
> > >
> > > &nbsp;
> > >
> > > &nbsp;
> > >
> > > [1] RT-1: Robotics transformer for real-world control at scale. Brohan et al. RSS. 2022.
> > >
> > > [2] On bringing robots home. Shafiullah et al. arXiv:2311.16098. 2023.
> > >
> > > [3] RoboCasa: Large-Scale Simulation of Everyday Tasks for Generalist Robots. Nasiriany et al. RSS. 2024.

---

### Note · Authors · 2025-08-12

We are delighted by the positive feedback from the reviewers and pleased that the new experiments and analyses we added further reinforce the recognized strengths of the paper and address any remaining concerns.

&nbsp;

### **Strengths recognized by all reviewers:**

- **Clear and compelling motivation** (Reviewers AqtN, XMz4)
- Introduces an **interesting and novel technical approach**, improving both **accuracy** and **efficiency** of video reasoning (Reviewers AqtN, Y85s, XMz4, JuQn)
- Includes **rigorous and comprehensive experiments** with consistent performance improvements (Reviewers AqtN, Y85s, XMz4, JuQn)
- Provides a **valuable public dataset contribution** (Reviewers AqtN, Y85s, JuQn)

&nbsp;

### **Concerns and how we resolved them:**

- Lack of decomposition failure analysis
  - We **add a comprehensive failure analysis** for both **real-world videos** and **videos from sim** (**Table R1**).
  - These results will be added in the main results section, with extended discussion in the appendix.

- Limited real-world validation
  - We **add extensive real-world experiments** (**Table R2**), testing ROVER on **the largest collection of real-world robotics datasets** publicly available (the OXE datasets).
  - In this analysis, ROVER shows **substantial improvements** over strong baselines in the diverse **real-world settings**.
  - These results will be added to the main results section.

- Extended baseline comparisons
  - We **add 3 more modern baselines** that are specifically designed for video understanding for our **real-world and sim evaluations** (**Tables R3, R4, R5**). We also add a baseline for local reasoning (**Tables R16, R17**).
  - In this analysis, ROVER continues to show **significant performance gains** over all baselines on all video reasoning and QA tasks.
  - These baselines will be added to Figures 3, 5, and 7 (main results).

- Potential dependence on task structure
  - We add more extensive evaluation to illustrate ROVER’s effective reasoning with **real-world tasks that do not have standard object-centric subtasks** (e.g., cable routing).
  - These results will be added to the main results section, with clear demonstrations in the appendix.

&nbsp;

### **Summary:**
The work’s **novelty, clear motivation, rigorous methodology, comprehensive evaluation, and strong results** are acknowledged across all reviewers. The remaining concerns of the reviewers have been fully addressed with new experiments and analyses.

---

### Decision · Program_Chairs · 2025-09-17

**Decision:**

Accept (poster)

**Comment:**

The paper proposes an approach for video reasoning in embodied settings to primarily address the limitations of VLMs in reasoning over long sequences of frames. The paper received divergent ratings: 2 Accept and 2 Borderline Reject. The reviewers mentioned various concerns such as: (1) Lack of analysis, (2) Limited set of modern baselines, (3) Limited real-world validation, (4) Lack of validation for linear time complexity, and (5) No testing inside a complete planning and execution loop. The rebuttal addressed most of the issues. For instance, the authors addressed (2) by providing new baseline results such as VideoGemini and addressed (4) by providing a new analysis.

After the discussions, there were some concerns such as being limited to only table-top validation. The AC checked the paper, the reviews and the rebuttal. The AC believes the positives of the paper outweighs its negatives, and the paper addresses an important problem with thorough experiments. Hence, acceptance is recommended. However, the authors are encouraged to explore applications beyond simple table-top scenarios and use the proposed method inside a complete robotic pipeline.